# Explainable Semantic Space by Grounding Language to Vision with Cross-Modal Contrastive Learning

**Yizhen Zhang**[1,2]**, Minkyu Choi**[1]**, Kuan Han**[1]**, and Zhongming Liu**[1,3]

[1] Department of Electrical Engineering and Computer Science, University of Michigan, Ann Arbor, MI 48109
[2] Department of Neurological Surgery, University of California San Francisco, San Francisco, CA 94143
[3] Department of Biomedical Engineering, University of Michigan, Ann Arbor, MI 48109
{zhyz, cminkyu, kuanhan, zmliu}@umich.edu

## Abstract

In natural language processing, most models try to learn semantic representations merely from texts. The learned representations encode the "distributional semantics" but fail to connect to any knowledge about the physical world. In contrast, humans learn language by grounding concepts in perception and action and the brain encodes "grounded semantics" for cognition. Inspired by this notion and recent work in vision-language learning, we design a two-stream model for grounding language learning in vision. The model includes a VGG-based visual stream and a Bert-based language stream. The two streams merge into a joint representational space. Through cross-modal contrastive learning, the model first learns to align visual and language representations with the MS COCO dataset. The model further learns to retrieve visual objects with language queries through a cross-modal attention module and to infer the visual relations between the retrieved objects through a bilinear operator with the Visual Genome dataset. After training, the model's language stream is a stand-alone language model capable of embedding concepts in a visually grounded semantic space. This semantic space manifests principal dimensions explainable with human intuition and neurobiological knowledge. Word embeddings in this semantic space are predictive of human-defined norms of semantic features and are segregated into perceptually distinctive clusters. Furthermore, the visually grounded language model also enables compositional language understanding based on visual knowledge and multimodal image search with queries based on images, texts, or their combinations.

## 1 Introduction

Humans take much longer time to name a colored word when the color and the word mismatch (e.g., "*red*" shown in green) than when they match (e.g., "*red*" shown in red) [1]. This effect is an example of rich psychological evidence suggesting that humans learn language by grounding meanings to knowledge about the world [2, 3]. In contrast, most models in natural language processing (NLP) [4–7] encode "distributional semantics" [8] learned from texts only. Put yourself as machines in a thought experiment for the "Chinese Room Argument" [9]. Imagine that you have to learn Chinese from scratch as your first language. All that you have is a Chinese-to-Chinese dictionary. You might be able to relate a word to other words based on textual distributions. It is, however, impossible to learn word meanings without any additional explanation in reference to the physical world [10].

A language model may learn concepts from texts paired with sensory data, such as images. Joint vision-language learning has been explored for image captioning [11], visual question answering [12], and pre-training vision models with weak supervision [13, 14]. In line with these studies, we

train a language model and a vision model jointly to match images and texts. We further analyze the semantic space obtained with the visually grounded language model. In this space, semantic embeddings are found to be organized and clustered by visual attributes, predictive of human-defined norms of semantic features, useful for compositional language understanding and cross-modal image search. We expect this visually grounded language model to also be useful for understanding the computational basis of grounded cognition [15, 16].

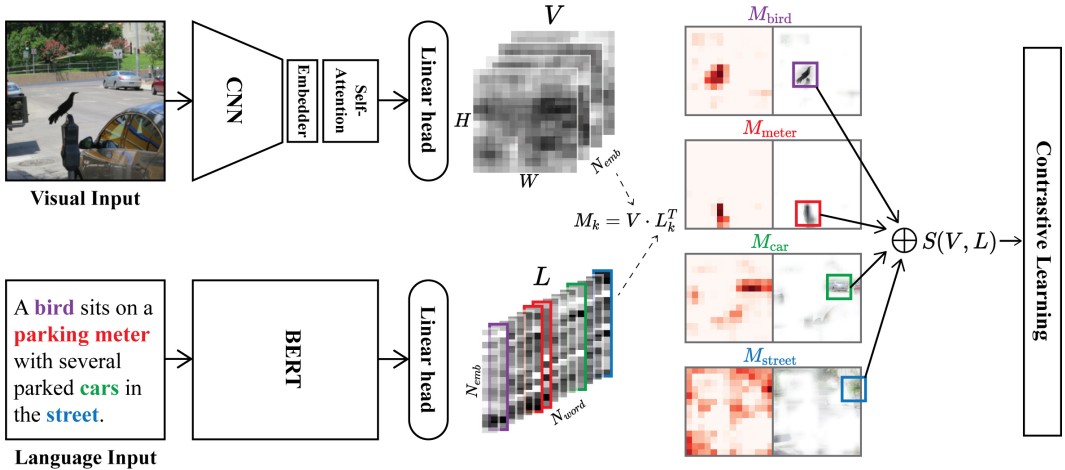

Figure 1: Visual grounding of natural language (see Section 3.1). The visual and language streams take an image and its caption as input, respectively. The inner-product between the visual feature maps and the contextual word embeddings forms the 3D match-map that highlights the matching between visual and language content. The similarity score calculated from the match-map (see Eq. 1) is used to evaluate the cross-modal contrastive loss.

## 2 Background and Related Work

### 2.1 Distributional vs. grounding hypothesis

In the *distributional hypothesis* [17], words that occur in similar contexts carry similar meanings. This hypothesis has motivated influential machine learning models to learn word embeddings from large text corpora [4, 18]. However, the learned word embeddings are not straightforward to interpret [8, 19]. Alternatively, the *symbol grounding hypothesis* suggests that a word is connected to its meaning by relating to its referent in the physical world [10, 20]. In line with this hypothesis, earlier studies demonstrate that visual features or contexts can enhance language learning [21–25].

### 2.2 Vision-language learning

Grounding language in vision has been of increasing interest in computational linguistics and machine learning. A common strategy is to fuse words with related visual information in terms of perceptual norms [21], bag-of-visual-word [23, 26], or learnable visual features [27–30]. The models used for vision-language fusion evolve alongside those for NLP, such as Latent Dirichlet Allocation (LDA) [23], log-bilinear model [31], Skip-gram model [25, 27], and recurrent neural network [28]. Generally, visual grounding may refine the distribution and interpretability of language representations [23, 25, 26, 28–32] and facilitate cross-modal tasks [28–33]. More recent work has begun to use transformer [6, 34] for vision-language learning, showing strong performance in cross-modal tasks [35–38] .

Contrastive learning [39, 40] is increasingly applied to not only unimodal data [41] but also multi-modal data [13, 42]. It is able to learn better representations than alternative prediction or classification objectives [43]. However, cross-modal contrastive learning is still under-explored for higher-level tasks, e.g., visual question answering [12], visual reasoning [44], scene graph generation [45]. Such tasks involve abstract reasoning about the relations between entities (e.g., visual objects). Prior

work approaches relational inference with multi-layer perceptron [46, 47] or graph neural networks [48–50]. Arguably, a more compelling idea [51] is to model entities as vectors in a continuous space and to model their relations as arithmetic operators (linear [52–54] or bilinear [55, 56]) applied to the vector representations of those entities.

## 2.3 Relation to prior work

In this work, we first build a two-stream model to jointly learn visual and language representation from image-caption pairs, similar to recent work [13, 14]. We then finetune the learned model by adding a cross-modal attention layer [35, 36] and bilinear operators [55] to represent the relations between visual objects. Both stages utilize cross-modal contrastive loss. Related to our work, Harwath et al. match visual objects to spoken words using triplet loss [42]. Early this year, Jia et al. [13] and Radford and Kim et al. [14] use contrastive learning to pretrain a vision model using a massive image-text dataset and demonstrate largely improved zero-shot transfer learning performance on visual and cross-modal tasks. Different from their perspectives, we focus on assessing the language encoders and word representations. Specifically, we perform a systematic evaluation of the semantic space grounded in vision vs. the ungrounded semantic space learned from texts only. This evaluation is possible since after training, the language and visual streams in our model are fully separable as stand-alone systems, unlike some vision-language models that require both visual and textual input to be usable [35, 36]. Our goal is to assess how visual grounding affects the distribution of textual representations by analyzing the distribution of word embeddings in the grounded semantic space, in line with related works [23, 25, 26, 28–32, 57] .

## 3 Approach

### 3.1 Visual grounding of natural language

To build a computational model for learning visually grounded language representations, we develop a model (Fig. 1) that combines a stand-alone visual stream and a stand-alone language stream. The visual stream is based on VGG16 [58] with an additional linear transformation as an embedder to match the feature dimension of the language stream and an additional multi-head self-attention layer [59] to enforce global information aggregation and learn long-range dependency. The language stream is based on Bert [6]. Using separate linear transformation heads [41], the output from both the visual stream and the language stream are projected to a common representational space. In this common space, the inner-product between the visual representation $V$ at every location and the language representation $L$ of every word gives rise to a 3D match-map, where each element indicates how a word in the text matches each location in the image (See illustration in Fig. 1). The sum of the maximal match is the similarity score $S(V, L)$ between a pair of image and text. See Eq. 1, where $i, j$ indicate the location in the 2D image feature map $V$ and $k$ indicates the $k$-th word in $L$.

$$M_{i,j,k} = V_{i,j} \cdot L_k^T, \quad S(V, L) = \sum_{k=1}^{K} \max_{i,j} M_{i,j,k} \tag{1}$$

Extending the unimodal normalized temperature-scaled cross-entropy (NT-Xent) loss [13, 41, 60], we define the cross-modal contrastive loss using the anchor sample from one modality and the positive sample and negative samples from the other modality [13, 14]. As such, we define and sum two loss functions with the anchor sample from either images or texts and positive/negative samples from either texts or images, respectively.

$$\text{Loss}_l = -\frac{1}{B} \sum_{i=1}^{B} \log \frac{\exp(S(V_i, L_i)/\tau)}{\sum_{j=1}^{B} \exp(S(V_i, L_j)/\tau)}, \quad \text{Loss}_v = -\frac{1}{B} \sum_{i=1}^{B} \log \frac{\exp(S(V_i, L_i)/\tau)}{\sum_{j=1}^{B} \exp(S(V_j, L_i)/\tau)} \tag{2}$$

For $\text{Loss}_l$ in Eq. 2, the anchor sample $V_i$ is an input image and the positive sample $L_i$ is the corresponding image caption, whereas the negative samples $L_j$ are unmatched textual descriptions included in the same batch ($B$ is the batch size). Similarly, $\text{Loss}_v$ in Eq. 2 is defined to contrast the positive and negative image samples against an anchor textual sample.

## 3.2 Visual grounding of object relations

We further finetune the model for visual relation prediction, as illustrated in Fig. 2. In this stage, we remove the linear transformation heads in Fig. 1 and add a multi-head cross-modal attention module [35, 36]. The attention module uses a query based on the embedding of an object word from the language stream (Query$_L$) and uses keys (Key$_V$) and values (Value$_V$) from every location in the feature map output from the visual stream. The attention score is calculated as the inner-product of Query$_L$ and Key$_V$ followed by softmax. The attention-weighted sum of Value$_V$ is concatenated across 8 attention heads to generate a visually grounded object representation.

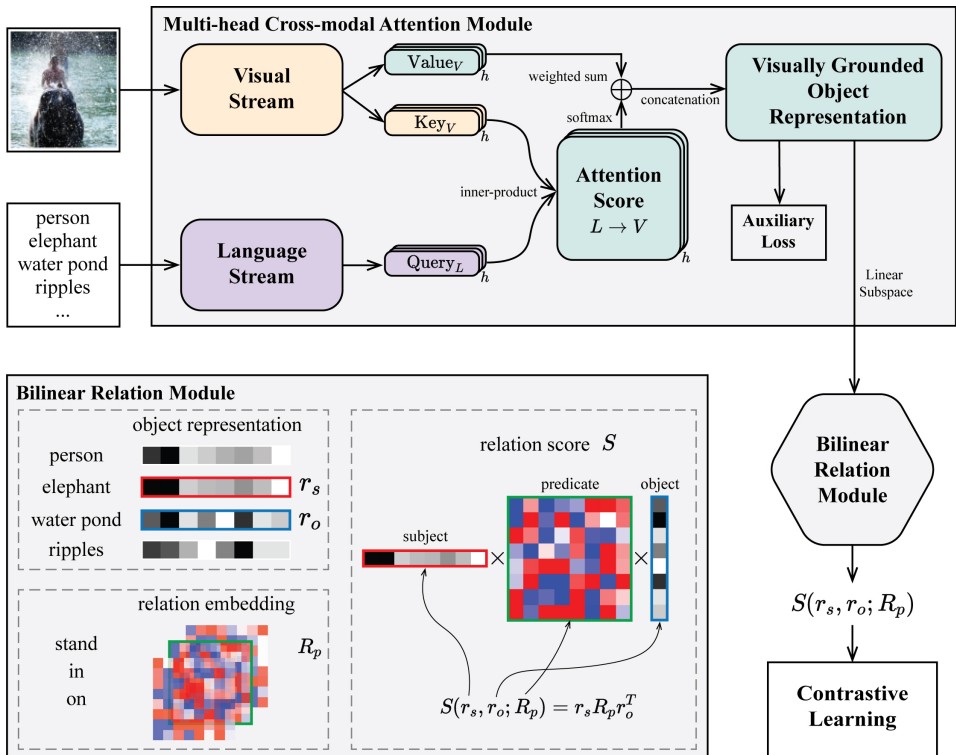

Figure 2: Visual grounding of object relation. The language stream uses an object description as input (e.g., `large black elephant`; we only show the object name "elephant" in this illustration for simplicity). The multi-head cross-attention module outputs a set of visually grounded object representations (See detailed methods in Appendix A.3). The bilinear relation module (bottom left) further generates a relation score given representations of a (**s**ubject,**p**redicate,**o**bject) triplet (e.g. (`elephant,in,water pond`)) for contrastive learning.

Applied to the grounded object representation is a bilinear relation module for predicting the visual relation between two objects (linguistically a subject and an object). For both the subject and the object, their grounded representations are linearly transformed to a subspace $D = \mathbb{R}^d$ (default $d = 32$), denoted as $r_s$ and $r_o$, respectively. A predicate $p$ is represented as a learnable bilinear operator $F_p : D \times D \rightarrow \mathbb{R}$, which represents the relation embedding $R_p$. Applying this bilinear operator to the subject vs. object representations measures their relation score $S$ specific to the given predicate [55] expressed as (Eq. 3).

$$S(r_s, r_o; R_p) = F_p(r_s, r_o) = r_s R_p r_o^T \tag{3}$$

For visual relation prediction, we also use contrastive learning with two loss functions by taking either relation embedding or subject/object representations as positive/negative samples.

$$\text{Loss}_{\text{rel}} = -\frac{1}{|\mathcal{B}|} \sum_{(r_s, r_o; R_p) \in \mathcal{B}} \log \frac{\exp(S(r_s, r_o; R_p)/\tau)}{\sum_{k \in \mathcal{K}_{\text{rel}}} \exp(S(r_s, r_o; R_p^k)/\tau)} \tag{4}$$

$$\text{Loss}_{\text{obj}} = -\frac{1}{|\mathcal{B}|} \sum_{(r_s, r_o; R_p) \in \mathcal{B}} \log \frac{\exp(S(r_s, r_o; R_p)/\tau)}{\sum_{k \in \mathcal{K}_{\text{obj}}} \exp(S(r_s^k, r_o^k; R_p)/\tau)} \tag{5}$$

In $\mathrm{Loss_{rel}}$, $\mathcal{K}_{rel}$ is the set that contains all relations available. The anchor sample is a pair of subject and object in an image. The positive sample is the embedding of the ground truth relation. The negative samples are the embeddings of all other relations. In $\mathrm{Loss_{obj}}$, the anchor sample is a given relation. The positive sample is a subject-object pair that holds this relation. The negative samples are other subject-object pairs in a different relation. For both loss functions, the positive and negative samples are drawn from the same batch $\mathcal{B}$.

In addition, we also add a classification head (two fully connected layers with ReLU in between) and apply it to the grounded object representation. We use object classification as an auxiliary objective (with a cross-entropy loss) to constrain the grounded object representation to be separable across objects for classification.

### 3.3 Training and Testing

We train the model in three stages to progressively refine the model with increasingly demanding tasks. In the first stage, we pretrain the visual and language streams separately as image and text encoders. The language stream is the pretrained Bert[1] used as the baseline model for subsequent experiments. The visual stream is pretrained for object classification with ImageNet [61]. Relative to the baseline CNN, the inclusion of self-attention improves the top-1 classification accuracy from $71.6\%$ to $74.3\%$ on the ImageNet validation dataset. The attention module also renders the classification more robust when the input image is partially occluded (See details in Appendix A.1).

In the second stage, we refine the pretrained language and visual streams by matching texts to images, as illustrated in Fig. 1 on the MS COCO dataset [11]. While freezing other layers, we refine the self-attention layer in the visual stream and the top $k$ layers in Bert (by default $k = 8$). Training with contrastive learning is based on the MS COCO dataset. As five captions are available for each image, we randomly sample one caption per image in each iteration. Earlier grounding (larger $k$) tends to support better image-text retrieval performance (see details in Appendix A.2).

In the third stage, we further finetune the model for visual relation prediction as illustrated in Fig. 2. We refine the visual self-attention layer and the higher $l$ layers in Bert (by default $l = 2$) based on the Visual Genome dataset [45] after cleaning the dataset to include $114$ relations and $55$ object classes in order to alleviate imbalanced data across different classes or relations (Appendix A.3). The training does not use any image annotation (e.g., bounding box) , which otherwise requires other models (e.g., object detection). Instead, we use cross-attention to retrieve visual objects from raw images given textual queries and learn the representations of the retrieved objects and their relations altogether. After training, the model predicts the object class with $97.71\%$ top-1 accuracy and predicts the relation label with $64.26\%$ top-1 accuracy (See details and examples in Appendix A.3).

## 4 Experiments

### 4.1 Principal components of grounded semantic representations

To evaluate the visually grounded semantic space, we use the language stream as a stand-alone model to extract the output representations of commonly used English words in the SemCat dataset ($9,197$ words; 100 word categories) [62]. Details about how word representations are extracted from the language stream are explained in Appendix B. We apply the principal component analysis to the representations of all the words studied here and examine the top components as the principal dimensions of the grounded semantic space.

Table 1: Correlation between the $1^{\text{st}}$ principal axis and human-rated word concreteness

| Group | Correlation (Pearson's r) | | |
| --- | --- | --- | --- |
| | Bert | Grounded | Relational Grounded |
| word-level | 0.1040 | 0.6615 | **0.6948** |
| category-level | 0.3538 | **0.8749** | 0.8001 |

---

[1] bert-base-uncased: https://huggingface.co/transformers/pretrained_models.html

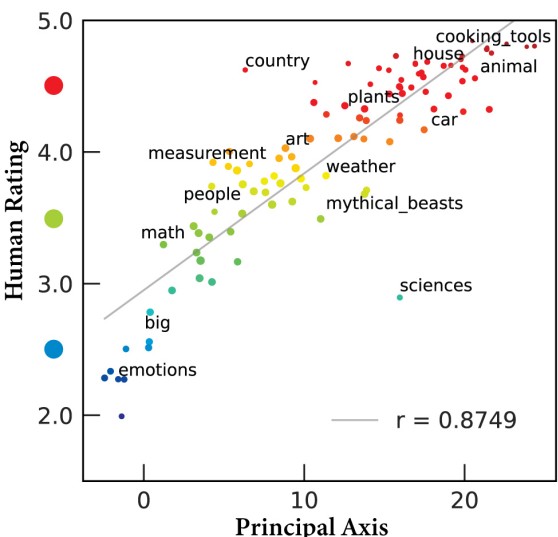

**Examples:**

| word | category | principal axis | human rating |
|---|---|---|---|
| oven | cooking tool | 30.86 | 4.97 |
| zebra | animal | 24.74 | 4.86 |
| car | car | 22.06 | 4.89 |
| furniture | house | 20.83 | 4.89 |
| defense | country | 4.65 | 4.19 |
| chemistry | sciences | 18.67 | 3.64 |
| wood | plants | 16.22 | 4.85 |
| cartoon | art | 11.84 | 4.33 |
| humid | weather | 11.58 | 3.48 |
| angel | mythical beasts | 7.22 | 3.82 |
| lover | people | 6.01 | 3.68 |
| thousand | math | 3.65 | 3.07 |
| huge | big | 0.17 | 3.54 |
| cheerful | emotions | - 2.54 | 2.34 |

Figure 3: The first principal component in the grounded semantic space captures the concrete-abstract axis of semantics. Left: Each dot represents a word category with the color indicative of the averaged human-rated concreteness (the y axis) and the size proportional to the standard deviation. The x axis indicates the value projected onto the first principal axis. Right: Example words in labeled categories.

Interestingly, the first principal dimension is readily interpretable as an abstract-to-concrete axis (Fig. 3). For example, words with the highest values in this axis are *ostrich, seagull, albatross, blender, pelican, broccoli, parakeet, lettuce, sailboat, vegetables*, whereas words with the lowest values are *displeasure, liking, to, outgoing, present, experienced, profitable, faithful, meaningful, multitude*. The representations of words along this axis is significantly correlated with human rating of their concreteness (ranging from 1 to 5) from prior study [63] (Fig. 3). The Pearson correlation coefficient reaches 0.8749 or 0.6615 across word categories or individual words, respectively; after grounding with object relations: $r = 0.8001$ for categories, $r = 0.6948$ for words. In contrast, the principal axis of the ungrounded semantic space learned from the baseline Bert model is not straightforward to interpret and shows a weak correlation with human ratings of concreteness (Table. 1). Other principal components are also intuitively interpretable. For example, PC 2 captures the human vs. non-human axis, PC 3 captures the scene vs. object axis, PC 4 captures the natural vs. artificial axis, PC 5 captures the indoor vs. outdoor axis, PC 6 highlights words related to food. See results about other principal components in Appendix B.1.

## 4.2 Relation to human-defined norms of semantic features

We further ask whether the visually grounded word embeddings are amenable to binary semantic features defined by humans [25, 64]. We use the concept property norm dataset from the Centre for Speech, Language and the Brain (CSLB) [65]. The dataset includes binary semantic features (e.g., `has_wheels`) labeled for 638 concepts collected from 123 human participants. We keep 390 features that each contains at least 5 samples. We hypothesize that the grounded word embeddings can be readout with a linear and sparse projection to readily support binary classification attainable by humans. To test this hypothesis, we train a logistic regression model with L1 regularization to predict each binary semantic feature from the grounded word embeddings and also repeat this for ungrounded semantics for comparison. See Appendix B.2 for details about this dataset and our evaluation method. Results suggest that the grounded word embeddings are significantly more predictive of visually relevant binary features than ungrounded counterparts obtained by Bert (Wilcoxon Signed Rank Test; $p < 0.0001$) (Fig. 4). This difference is less pronounced but still significant for other features related to other perceptual (e.g., `has_flavors`), functional (e.g., `does_cut`), encyclopaedic (e.g., `is_dangerous`), and taxonomic features (e.g., `is_clothing`), especially after visual grounding of object relations.

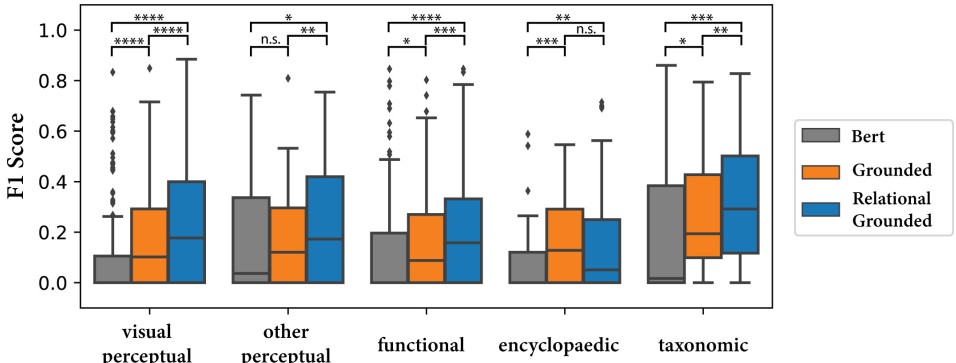

Figure 4: The F1 score of predicting semantic feature norms from word representations before and after visual grounding. Each box shows the lower (25%) percentile, the higher (75%) percentile, and the median of F1 scores within a feature type. Whisker= 1.5. Significant level: n.s.: not significant; *, $p < 0.05$; **, $p < 0.01$; ***, $p < 0.001$; ****, $p < 0.0001$.

## 4.3 Clustering of word representations

After visual grounding, the semantic representations tend to group themselves based on perceptual similarity. We use the SemCat dataset (9,197 English words from $N = 100$ categories) [62] and calculate the Silhouette coefficient (between $-1$ and 1) to measure the degree to which these words are clustered by categories. The distance between word embeddings is measured as the cosine distance (See details in Appendix B.3). The Silhouette coefficients across 100 categories are significantly higher for the visually grounded semantics than ungrounded ones (Wilcoxon Signed Rank Test; $p < 0.0001$) (Fig. 5 left). The greatest gain in clustering are noticeable for categories that include concrete concepts (e.g. `car`, `housing`, `mammal`) with defining visual attributes (Fig. 5 right). For some abstract categories related to human emotion (e.g., `happy`), the grounded representations are also better clustered than the ungrounded ones.

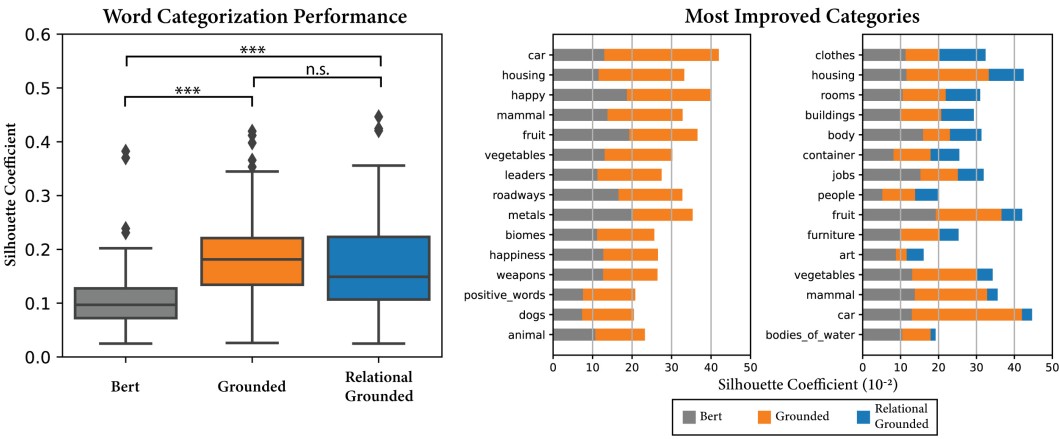

Figure 5: Left: A boxplot showing Silhouette coefficients on word representations before and after visual grounding. Each box shows the lower (25%) percentile, the higher (75%) percentile, and the median of the Silhouette coefficients. Whisker= 1.5. Right: Top-15 word categories that are better clustered after visual grounding of natural language and object relations.

## 4.4 Visually informed compositional reasoning

A drawback of distributional semantics is the inability to make visually informed compositional reasoning. We know that "zebra is a horse with black and white stripes", because we have seen how zebra looks like, whereas an ungrounded language model is never or rarely exposed to such

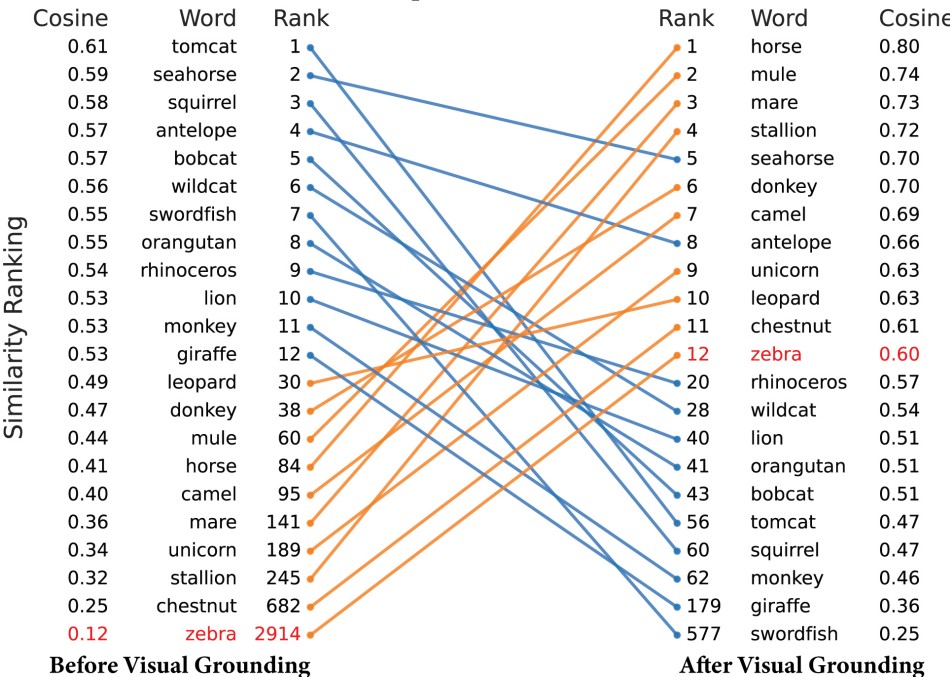

Figure 6: Compositional reasoning (**striped horse**). The left part shows the cosine similarity and its ranking between each of the listed words and the query phrase (striped horse) before visual grounding. The right part shows the corresponding results after visual grounding of natural language. Orange lines indicate words with increased ranking after visual grounding and blue lines for the decreased cases. We highlight in red the target word "zebra" for this specific example, which shows a significant increase in cosine similarity (from $0.12$ to $0.60$) and ranking (from $2914$ to $12$ out of $6238$ unique words). Besides, the top words similar to "striped horse" are all horse-like animals after visual grounding, but this is not the case for the ungrounded Bert model.

information [10]. We test whether the visually grounded semantics can perform compositional reasoning based on visual knowledge, without being explicit trained to do so. We choose some words (Table 2), for which the meaning can be intuitively inferred from the combination of other words.

Table 2: Examples of visually informed conceptual composition. Each row shows the cosine similarity and its ranking in the vocabulary (unique words in the Semcat dataset) between the query phrase and the target word. Except (`hot weather`, `summer`), all others are concepts supported by composition of *visual* knowledge in the query phrase. For each case, the highest similarity are rank are in bold.

| Query Phrase | Target Word | Similarity (cosine \| rank) | | | | | |
|---|---|---|---|---|---|---|---|
| | | Bert | | Grounded | | Relational | |
| striped horse | zebra | 0.12 | 2914 | 0.60 | 12 | **0.63** | **8** |
| black and white bear | panda | 0.13 | 2478 | 0.69 | **2** | **0.81** | **2** |
| flying car | plane | 0.36 | 167 | **0.66** | **4** | 0.61 | 11 |
| round container | bowl | 0.25 | 489 | 0.56 | 8 | **0.67** | **2** |
| red fruit | strawberry | 0.39 | 239 | 0.75 | **3** | **0.85** | **3** |
| young dog | puppy | 0.40 | 94 | 0.92 | **2** | **0.93** | **2** |
| iced mountain | glacier | 0.44 | 20 | **0.86** | **1** | 0.73 | 5 |
| clear sky | sunny | 0.27 | 631 | 0.31 | 184 | **0.34** | **61** |
| hot weather | summer | 0.27 | 903 | 0.52 | 14 | **0.53** | **6** |

For example, we use a phrase "striped horse" as a compositional query to search for the matched words ranked in terms of cosine similarity. In Fig. 6, the left part shows the cosine similarity and

ranking between each of the listed words and the query phrase "striped horse" before visual grounding. The right part shows the corresponding results after visual grounding of natural language. With the grounded semantic representation, the phrase `striped horse` is highly similar to the word `zebra` (cosine similarity: $0.60$), which is ranked as the $12$-th in the vocabulary. After further grounding the language model with visual object relations, the target word `zebra` has an even higher cosine similarity of $0.63$ ranked the $8$-th in the vocabulary (Table 2). Other top-ranked words all refer to horse-like animals (i.e., *horse, mule, mare, stallion, donkey, camel, antelope*). This is in sharp contrast to the ungrounded semantic space, in which it is impossible to relate `striped horse` to `zebra` based on the similarity of their representations (cosine similarity: $0.12$; rank: $2,914$). The ungrounded Bert model highlights the top-3 similar words as *tomcat, seahorse, squirrel*, which are animals sharing fewer visual features with horse-like animals. See other examples in (Table 2 and Appendix B.4).

### 4.5 Multimodal image search

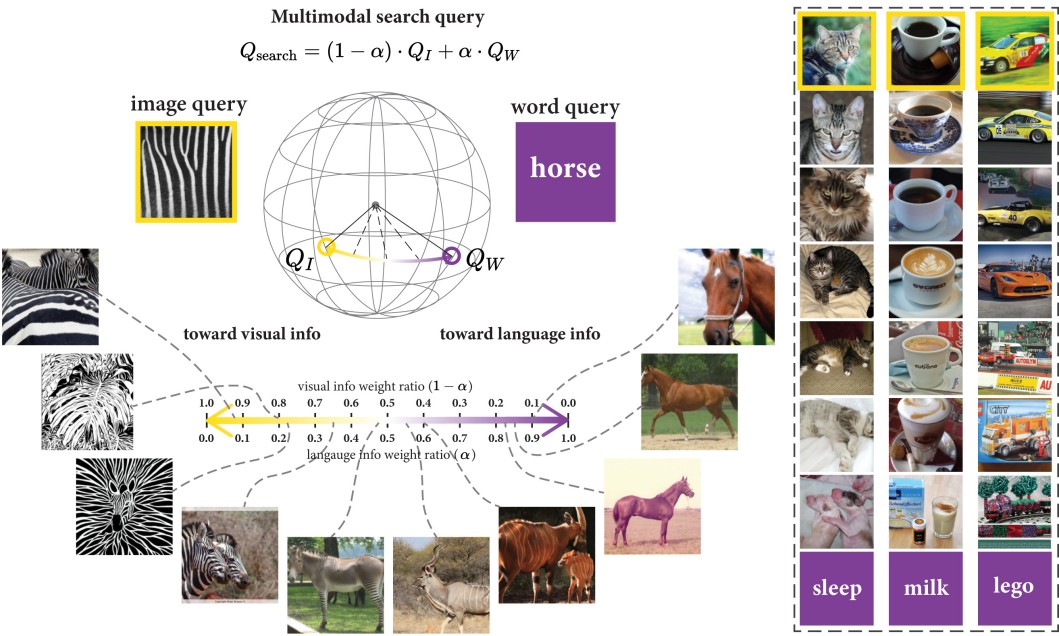

Figure 7: Left: Illustration of multimodal image search with a "zebra" example. The image query $Q_I$ is the L2-normalized vector representation of a zebra's skin pattern. The word query $Q_W$ is the L2-normalized vector representation of word *horse*. As the weight ratio $\alpha$ in multimodal query $Q_{\text{search}}$ increases from $0$ to $1$, the search results show progressive changes from stripped patterns, to a real zebra, and to a horse image. Right: Example multimodal image search results using other image-word pairs as the combined query. The query images (highlighted with a yellow boundary) are in the top row and the corresponding words (shown in a purple background) are in the bottom row. The 1st to the 3rd columns correspond to combined queries, with a "cat" image and a *sleep* word, a "coffee" image and a *milk* word, a "car" image and a *lego* word. The results of multimodal image search are shown in the 2nd to 6th rows, corresponding to increasing $\alpha$ from $0$ to $1$.

In our model, the cross-attention module forms a joint representational space to combine both visual and textual input. We explore whether this joint space can be used to support cross-modal tasks, e.g., image search based on image, text, or their combinations [13]. For this task, we add two additional heads ($F_V$ and $F_L$). Each head includes two linear layers with ReLU in between followed by average pooling (See details in Appendix B.5). It is applied to either visual or textual representations in the joint space and results in a single vector representation for an image or a text (Eq. 6, $d = 768$). While freezing our model described in Section 3.2, we train the two additional heads with contrastive loss to match the average-pooled representations of paired images and texts in terms of their cosine similarity using the MS COCO dataset. To use the model for image search, we apply weighted sum to

the normalized representations of a query image and a query text (with weights: $1 - \alpha$ and $\alpha$, where $0 \leq \alpha \leq 1$). We use this multimodal query (Eq. 7) to search a held-out database[2] for the matched images ranked in terms of cosine similarity.

$$\boldsymbol{Q}_I = F_V(\text{Key}_V) \in \mathbb{R}^d, \quad \boldsymbol{Q}_W = F_L(\text{Query}_L) \in \mathbb{R}^d. \tag{6}$$

$$\boldsymbol{Q}_{\text{search}} = (1 - \alpha)\boldsymbol{Q}_I + \alpha\boldsymbol{Q}_W. \tag{7}$$

As $\alpha$ controls the weighting between the textual and visual queries, we test how the image search returns different results as $\alpha$ increases from $0$ (image only) to $1$ (text only). For example, when we combine a word (*horse*) and an image (a stripped pattern) into a query, the search finds images similar to the zebra's skin pattern when $\alpha$ is close to $0$, or finds images of typical horses when $\alpha$ is close to $1$, but not necessarily a zebra for either case until when $\alpha$ is somewhere close to $0.5$ (Fig. 7, left). This observation is generalizable to other examples. See similarly graded changes in (Fig. 7, right).

## 5  Discussion

In summary, we apply visual grounding to not only words but also relations between words through cross-modal contrastive learning. The results suggest that grounding language learning in vision renders semantic representations more interpretable by human intuition. The grounded semantic space has its principal dimension encode the concrete-to-abstract variation consistent with human ratings and neurobiological knowledge. The grounded semantic representations are better clustered by finer categories and capable of compositional reasoning (e.g., `zebra = striped horse`). In addition, our work also shows compelling evidence that both text and image-informed semantics are represented in a common, continuous, and grounded semantic space. Although this notion has been hypothesized in neuroscience and linguistics, it has been rarely implemented and demonstrated with computational models. Uniquely, we demonstrate that a continuously varying combination of a text and an image into a multimodal query can be used to search images, showing results that make intuitive sense.

Several limitations of our work are noteworthy. The datasets used to train our model are orders of magnitude smaller than those used in recent studies [13, 14]. Scaling up the model training with increasingly larger datasets is expected to greatly improve the model's performance for cross-modal tasks, while generally preserving the interpretability of the grounded language representations as described herein. Some of our experiments and results are preliminary and primarily for illustrative purposes and await more comprehensive and quantitative evaluation in future studies, especially with more downstream vision-language tasks. Whereas our evaluation focuses on the language model, grounding language to vision may also have refined the visual stream, awaiting further evaluation against visual tasks, as demonstrated in [13, 14].

The visually grounded language model may be usable as a computational model for studying the grounded cognition - a theory in cognitive science [15, 16]. Ungrounded linguistic models are explanatory about semantic processing in the brain's language network [66–70]. Combining the grounded language model with human behavioral and neural data may elucidate how the language network interacts with distributed sensory and motor areas for semantic processing [71].

It is natural to extend this study by incorporating other sensory input [72, 73] and further ground language learning in action [74–77] and emotion [78]. This study also leaves an open question as to whether grounding should occur at an early or late stage of natural language processing, which awaits further exploration and evaluation. For comprehensive modeling of language grounding, it is desirable to expose an agent to a naturalistic and multi-sensory environment and to engage interactive actions to allow the agent to learn knowledge in the physical world, like how humans learn language.

## Acknowledgments and Disclosure of Funding

Funding in direct support of this work: NSF IIS 2112773.

---

[2] $41,600$ images from the validation dataset of Open Images Dataset $V6$.

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
