# Appendix: Explainable Semantic Space by Grounding Language to Vision with Cross-Modal Contrastive Learning

**Yizhen Zhang**[1,2], **Minkyu Choi**[1], **Kuan Han**[1], **and Zhongming Liu**[1,3]

[1] Department of Electrical Engineering and Computer Science, University of Michigan, Ann Arbor, MI 48109
[2] Department of Neurological Surgery, University of California San Francisco, San Francisco, CA 94143
[3] Department of Biomedical Engineering, University of Michigan, Ann Arbor, MI 48109
{zhyz, cminkyu, kuanhan, zmliu}@umich.edu

## A    Training and Testing

### A.1    Visual stream pretraining

#### A.1.1    Training for ImageNet classification

We pretrain the visual stream on ImageNet [1] for object classification to evaluate whether adding a self-attention layer to VGG16 can help learn a better image representation. The linear embedder transforms the image feature from $512$ channels to $768$ channels to match the output feature dimension in the Bert model. This linear embedder also prepares the image and language representations for them to be merged in the second stage (see section A.2). In the visual stream, the single-layer self-attention has 12 heads and follows the structure as described in the original transformer paper [2]. We add a 2D positional encoding $\text{PE}(x, y)$ to the input feature map before passing it to the self-attention layer. The positional encoding is learnable and includes $768 \times 14 \times 14$ parameters. We use the same hyper-parameter setting for training VGG16 and attention-enhanced visual stream (batch size$= 200$, optimizer$=$SGD, learning rate$= 0.01$, momentum$= 0.9$, weight decay$= 1e-4$; learning rate decay by half for every 20 epochs). The training is done with $4$ Nvidia GeForce GTX Titan XP Graphic Card (12 GB memory per card).

The performance on ImageNet classification (see Table 1) is compared between VGG16 and its variation with attention enhancement. This result suggests that adding one additional self-attention layer on the top of VGG16 improves the classification performance on ImageNet.

Table 1: Object classification accuracy on ImageNet validation dataset.

| Model | Object classification accuracy (%) | | |
| --- | --- | --- | --- |
| | Top-1 | Top-5 | Top-10 |
| VGG16 | 71.6 | 90.4 | 94.0 |
| VGG16+attention | **74.3** | **91.8** | **95.1** |

35th Conference on Neural Information Processing Systems (NeurIPS 2021).

### A.1.2 Occlusion experiments

To evaluate how self-attention changes the feature representation, we further perform an occlusion experiment [3]. For each image in the validation dataset, a fixed-sized window ($32 \times 32$) centered at a specific location is occluded with a grey square. The center of this occlusion is iterated throughout the whole image (stride = 8) for individual trials of the occlusion experiment. Each trial of occlusion outputs a probability of the correct class. It is expected that after occluding different portions of the input image, the model prediction (i.e., the probability of classifying the occluded input as the correct label) may result in different confidence levels. If the occluded region includes a key feature of the correct class, the probability may drop significantly. The effect of occlusion is evaluated and visualized as a heat map, which shows the probability of correct classification as a function of the center of occlusion. For example (see Fig. 1), VGG16 fails to classify the image of jay (a bird) as the correct label when any part of the bird is occluded. After adding the self-attention layer, the classification is compromised only when a very small part of the image is occluded. Similarly, in the image of a bridegroom, the classification performance drops only when a key feature (the Boutonnière) is occluded, whereas the performance of VGG16 is sensitive to occlusions at multiple spots. In another example image (Newfoundland dog), the attention-enhanced model is insensitive to the occlusion placed anywhere. In rarer cases, attention makes the model more sensitive to occlusion. See the last row of Fig. 1. Such cases usually involve a large-sized object in the image and the object identity is most defined by the local texture (e.g., the dishcloth). Overall, adding the self-attention helps aggregate information across the image and makes the model much less sensitive to image occlusion.

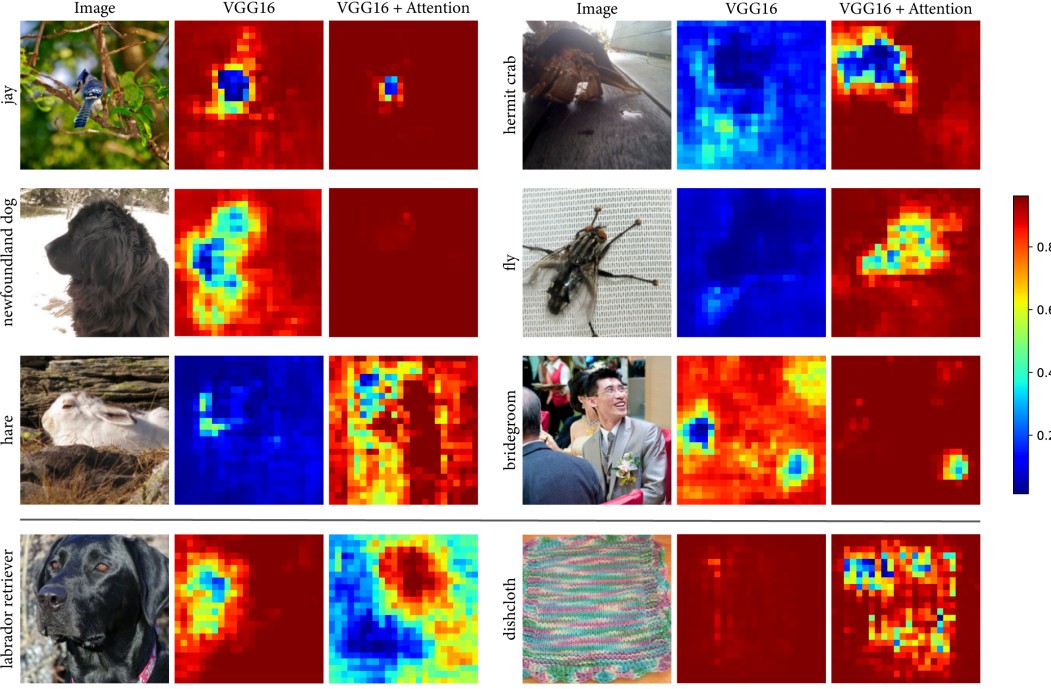

Figure 1: Example results from the occlusion experiment. Each example contains three images (from left to right): the input image, the heatmap showing the probability of correct classification by VGG16 given occlusion applied to different locations in the image, and the heatmap after adding the attention. The ImageNet class label is shown on the left. The first three rows show examples of when attention makes the model's performance less sensitive to occlusion. The last row shows examples of the opposite.

Quantitatively, we compare the probability of correct classification between VGG16 and its variation with attention for each trial of occlusion. We count the number of trials that the attention mechanism increases (or decreases) the probability of correct classification relative to VGG16, and evaluate the histogram by the size of increase (or decrease). As shown in Fig. 2, visual attention improves

the classification of occluded images in many more trials than its baseline VGG16. Overall, the self-attention layer makes the model more robust against occlusion.

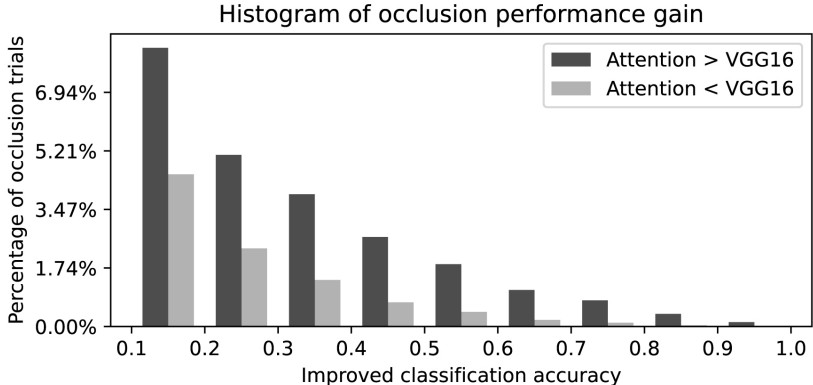

Figure 2: Quantitative results with occlusion placed at different parts of the input image in different trials. The $y$ axis shows the percentage of trials in which the attention-enhanced model shows better (dark grey) or worse (light grey) performance than VGG16. The $x$ axis shows the absolute difference in the probability of correct classification between the model with and without attention.

## A.2 Visual grounding of natural language with MS COCO

We pretrain the two-stream model with cross-modal contrastive learning on MS COCO dataset [4] as described in the main text Section 3.1. The training dataset consists of $118287$ images, each having $5$ captions. For each epoch, we randomly choose $1$ out of the $5$ captions. At this training stage, we freeze the convolutional layers (i.e., VGG16) in the visual stream and lower layers of the Bert encoder in the language stream.

For each self-attention layer in Bert, we also freeze the weights in query and key transformations (both are linear layers). This is motivated by two reasons. First, we want to control the number of learnable parameters to avoid over-fitting. Second, we want to separate the functional role of query ($Q$), key ($K$), and value ($V$) in self-attention. $Q$s and $K$s are trained to learn the syntactic and contextual relation between words in a sentence and $V$s are trained to learn word meanings. While $Q$s and $K$s have been trained adequately in the pretrained Bert, we freeze them to maintain the learned syntactic relations. We focus on refining $V$s in order to learn and represent word meanings in reference to visual perception, while leveraging both *textual* context and *multimodal* context.

We train the model with `Adam` optimizer (learning rate$= 5e-5$, weight decay$= 5e-7$, $\beta = (0.95, 0.999)$; dropout$= 0.3$; learning rate decay by half after every $15$ epochs; batch size$=180$; total training epochs$=100$). The temperature parameter in the contrastive loss is always set to $0.1$. The parallel training is done with $3$ Nvidia GeForce RTX 2080 Ti Graphics Card (each card with $11$ GB memory).

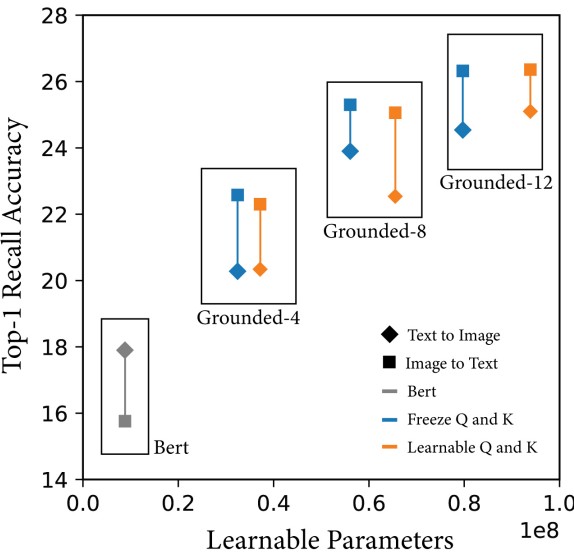

Figure 3: Cross-modal retrieval performance on MS COCO. The $x$ axis refers to the number of learnable parameters at this training stage. The label under the black box refers to a "grounded" language stream with the top $4$, $8$, or $12$ layers learnable, while the rest layers are fixed as are pretrained. `Bert`: the whole language stream is frozen; `Grounded-k`: the top $k$ layers in Bert are learnable.

Fig. 3 shows the image-to-text and text-to-image retrieval performance on the MS COCO validation set, which contains $5000$ images. The results suggest that making more layers in Bert learnable, which is interpreted as earlier stage of visual grounding, tends to result in better cross-modal retrieval accuracy, while freezing weights on query and key transformations (blue dots) can reduce the number of learnable parameters without compromising the performance.

An ablation study on contrastive losses further suggests that both $\text{Loss}_l$ and $\text{Loss}_v$ (as defined in Equation 2 in the main text) are important for this stage of training (Table 2). In brief, if we only use $\text{Loss}_l$ (which contrasts between positive and negative language keys), the text-to-image retrieval drops by $6.5\%$, while the image-to-text retrieval just increases by $0.2\%$. On the other hand, if we only use $\text{Loss}_v$ (which contrasts between positive and negative image keys) for training, the image-to-text

retrieval significantly drops from $25.3\%$ to $1.4\%$, although the text-to-image retrieval performance increases by $1.6\%$. The overall cross-modal retrieval achieves the best performance when both losses are used for model training.

Table 2: Ablation study of cross-modal contrastive losses (tested on Grounded-8 model).

| Loss function | Cross-modal retrieval accuracy (%) | |
| --- | --- | --- |
| | image-to-text | text-to-image |
| $\text{Loss}_l + \text{Loss}_v$ | 25.3 | 23.9 |
| $\text{Loss}_l$ | 25.5 | 17.4 |
| $\text{Loss}_v$ | 1.4 | 25.5 |

### A.3 Visual grounding of object relations with Visual Genome

#### A.3.1 Details in cross-modal attention and bilinear relational modules

For each head in the cross-modal attention module, the queries ($\text{Query}_L$) are from the object descriptions encoded by the language stream; the keys ($\text{Key}_V$) are from visual features in the visual stream (Eq. 1). The attention score $\boldsymbol{A}_{L \to V}$ is the inner-product between $\text{Query}_L$ and $\text{Key}_V$ (Eq. 2). The attention-weighted sum of the $\text{Value}_V$ from the visual stream (Eq. 1) is concatenated across different heads to generate a visually grounded object representation (Eq. 3).

$$\text{Key}_V^i = \boldsymbol{V}\boldsymbol{W}_K^i, \quad \text{Value}_V^i = \boldsymbol{V}\boldsymbol{W}_V^i, \quad \text{Query}_L^i = \boldsymbol{L}\boldsymbol{W}_Q^i \tag{1}$$

$$\boldsymbol{A}_{L \to V}^i = \text{softmax}\{\text{Query}_L^i(\text{Key}_V^i)^T / \sqrt{d}\}, \tag{2}$$

$$\boldsymbol{O} = \text{concat}\{\boldsymbol{A}_{L \to V}^1 \text{Value}_V^1, \cdots, \boldsymbol{A}_{L \to V}^h \text{Value}_V^h\} \tag{3}$$

where $i$ in Eq. 1 and Eq. 2 refers to the $i$-th attention head. $d$ in Eq. 2 refers to the query/key feature dimension. $h$ in Eq. 3 refers to the total number of attention heads (by default $h = 8$).

In the bilinear relational module, each predicate is represented by a matrix $\boldsymbol{R}_p \in \mathbb{R}^{d \times d}$ that encodes a specific relation between two visual objects. The relation score defined in Eq. 4 measures to how well the predicate $p$ describes the relation between a subject $s$ and a object $o$. We constrain the Frobenius norm (Eq. 5) to each relational embedding matrix such that the compositionality of relations follows the transitivity property. That is, the representation of the optimal relation between the object 1 and object 3 is the multiplication of the relation between object 1 and object 2 and the relation between object 2 and object 3 (Eq. 6):

$$S(\boldsymbol{r}_s, \boldsymbol{r}_o; \boldsymbol{R}_p) = F_p(\boldsymbol{r}_s, \boldsymbol{r}_o) = \boldsymbol{r}_s \boldsymbol{R}_p \boldsymbol{r}_o^T \tag{4}$$

$$\boldsymbol{R}_{(\boldsymbol{r}_1, \boldsymbol{r}_2)}^* = \underset{\|\boldsymbol{R}\|_F = 1}{\arg\max}\, S(\boldsymbol{r}_1, \boldsymbol{r}_2; \boldsymbol{R}) = \frac{\boldsymbol{r}_1^T \boldsymbol{r}_2}{\|\boldsymbol{r}_1\|_2 \|\boldsymbol{r}_2\|_2} \tag{5}$$

$$\boldsymbol{R}_{(\boldsymbol{r}_1, \boldsymbol{r}_3)}^* = \boldsymbol{R}_{(\boldsymbol{r}_1, \boldsymbol{r}_2)}^* \boldsymbol{R}_{(\boldsymbol{r}_2, \boldsymbol{r}_3)}^* \tag{6}$$

#### A.3.2 Data cleaning for relation and object labels and training parameters.

Since the relation labels are imbalanced in the original Visual Genome dataset [5], we filter the data to create a cleaner dataset for training and testing the proposed model to perform visual relation prediction task as well as the object classification, which is used as an auxiliary learning objective. To define object labels, we first extract the WordNet [6] synset for each object in Visual Genome data annotations. We investigate the distribution of the hypernyms of all object synsets and summarize them into 55 general classes as shown in Table 3. To define relation labels, we first extract the "predicate" term for each pair of objects in the data annotations and only preserve the ones with more than 250 instances in the Visual Genome dataset (which remains 292 out of 37342 unique labels). We then manually merge equivalent predicates into a single relation label (e.g., merge "*near*, *next to*, *on side of*, *beside*, *standing next to*, *next*, *standing near*, *to right of*, *near a*, *close to*, *on side*, etc." to "**near**"). In this way, we end up with 114 unique relation labels as shown in Table 4. Furthermore, we remove image samples with fewer than 5 subject-predicate-object triplets and keep a total of 98512 images. We then randomly split this cleaned dataset into training (93512 samples) and testing (5000 samples) set.

At this stage of training, we also freeze the CNN (the VGG16 encoder) in the visual stream and lower layers in the language stream, only keeping top 2 layers in Bert learnable. We train the model with `Adam` optimizer (learning rate= $1e-5$, weight decay= $5e-7$, $\beta = (0.95, 0.999)$; dropout= $0.1$; learning rate decay by half after every 15 epochs; batch size=180; total training epochs=150). The temperature parameter in the contrastive loss is always set to 1.0. The parallel training is done with 3 Nvidia GeForce RTX 2080 Ti Graphics Card (each card with 11 GB memory).

Table 3: Object labels (defined by WordNet synsets) for visually grounded object classification.

| | | |
|---|---|---|
| feline.n.0 | equine.n.01 | mammal.n.01 |
| bird.n.01 | animal.n.01 | body_part.n.01 |
| bread.n.01 | vegetable.n.01 | fruit.n.01 |
| meat.n.01 | beverage.n.01 | food.n.01 |
| tree.n.01 | herb.n.01 | vessel.n.02 |
| wheeled_vehicle.n.01 | aircraft.n.01 | vehicle.n.01 |
| road.n.01 | clothing.n.01 | furniture.n.01 |
| tableware.n.01 | home_appliance.n.01 | stairs.n.01 |
| building_material.n.01 | decoration.n.01 | room.n.01 |
| building.n.01 | container.n.01 | surface.n.01 |
| machine.n.01 | measuring_instrument.n.01 | instrument.n.01 |
| tool.n.01 | device.n.01 | paper.n.01 |
| man.n.01 | woman.n.01 | person.n.01 |
| equipment.n.01 | sport.n.01 | activity.n.01 |
| symbol.n.01 | sign.n.02 | number.n.02 |
| writing.n.02 | body_of_water.n.01 | facility.n.01 |
| geological_formation.n.01 | location.n.01 | atmospheric_phenomenon.n.01 |
| phenomenon.n.01 | communication.n.02 | structure.n.01 |
| artifact.n.01 | | |

Table 4: Relation labels for visual relation prediction task.

| | | | | |
|---|---|---|---|---|
| on | have | in | of | wear |
| with | behind | hold | near | under |
| by | above | sit | in front of | to |
| at | over | for | around | ride |
| stand | hang | carry | eat | walk |
| cover | play | lay | along | among |
| and | watch | belong to | painted | against |
| from | parked | made of | say | covered |
| mounted | across | fly | lying | grow |
| use | outside | cross | worn | printed |
| full of | filled with | swing | built | pull |
| touch | adorn | a | hit | support |
| written | lean | drive | rest on | held |
| connected to | cut | throw | line | through |
| float | show | face | graze | cast |
| stick out of | catch | drink | reflected in | be |
| beyond | lead | read | swim | white |
| off | seen | push | shining on | ski |
| wait | surf | down | make | feed |
| run | take | enjoy | that | at end of |
| stuck | reflect | stacked | black | plugged |
| overlook | form | without | do | kick |
| visible on | brush | blue | work on | |

### A.3.3 Ablation study on loss functions

Since the learning objective includes three loss functions ($\text{Loss}_{rel}$, $\text{Loss}_{obj}$, the auxiliary loss for object classification as described in Section 3.2). We also check how each loss function contributes to the model performance. We perform an ablation experiment by excluding out one loss at a time. The results suggest that the model has the best performance on relation prediction by combining all three losses (Table 5, Fig. 4). Whereas other losses do not appear to make a major difference, if $\text{Loss}_{rel}$ is excluded, the model would have much worse performance. This result suggests that $\text{Loss}_{rel}$ is the key component that allows the model to learn visual relation. We also find that $\text{Loss}_{obj}$ and $\text{Loss}_{rel}$ are somewhat entangled (i.e., minimizing one loss tends to decrease the other loss, see Fig. 5), but the model learns faster (Fig. 4) if we combine both two contrastive losses.

Table 5: Testing performance on ablation study of loss functions.

| Model | Object Classification | Relation Prediction (Top-1) | Relation Prediction (Top-10) |
|---|---|---|---|
| Combined loss | 97.71 | **64.26** | **95.21** |
| No auxiliary loss | 0.60 | 64.19 | 94.99 |
| No $\text{Loss}_{rel}$ | 97.95 | 29.97 | 68.78 |
| No $\text{Loss}_{obj}$ | **98.39** | 64.14 | 95.14 |

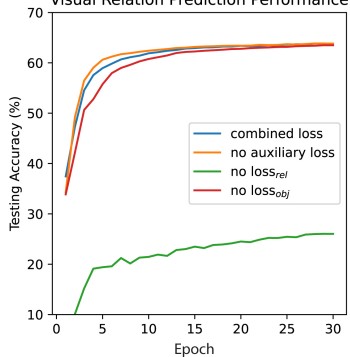

Figure 4: Learning curves for visual relation prediction on the testing dataset for the first 30 training epochs. The blue curve shows the performance when the model is trained with the loss that combines $\text{Loss}_{rel}$, $\text{Loss}_{obj}$, and the auxiliary loss for object classification, which is the default setting as mentioned in the main text of this work. The other three curves show the performance when the model is trained by excluding one of the three losses, as indicated in the figure legend.

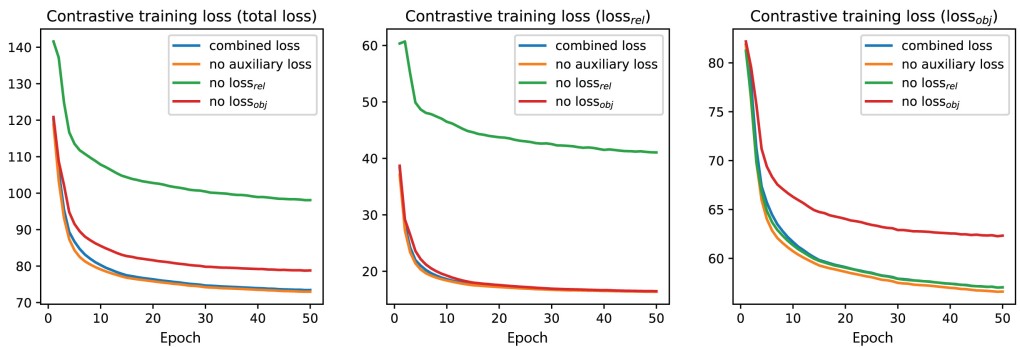

Figure 5: Learning curve of the contrastive loss functions. The *total loss* in the first figure refers to the summation $\text{Loss}_{rel} + \text{Loss}_{obj}$.

### A.3.4 Examples on testing dataset

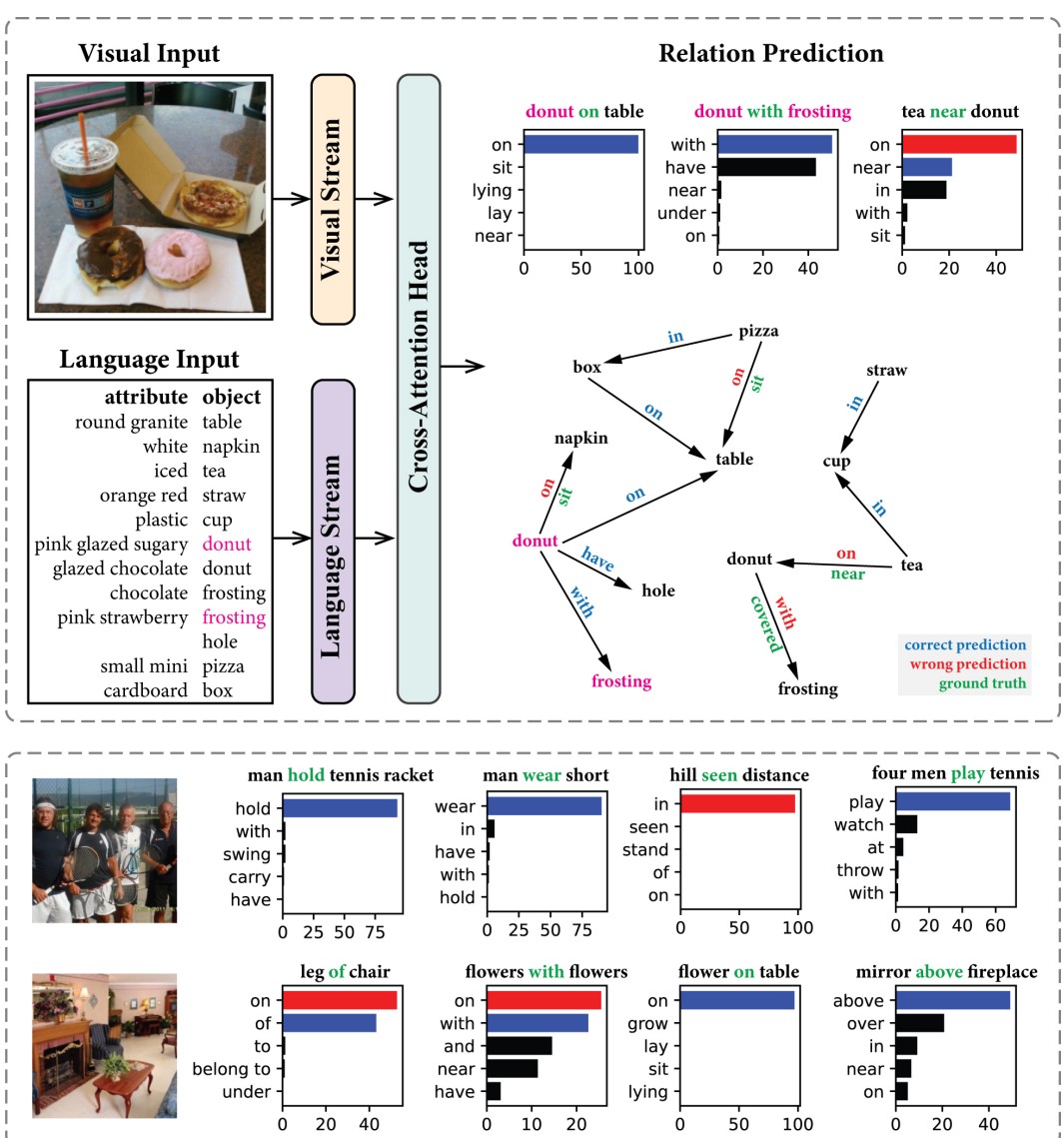

Figure 6: Examples of visual relation prediction task performance on testing dataset. Top: For visual grounding of object relations, the visual input is a natural image and the language input is a set of object descriptions. The bar charts show the examples for top-5 predicted visual relation of a paired objects. The directed graph shows top-1 predicted visual relations on all object pairs with ground truth labels in this example. Bottom: Other examples in the testing dataset.

# B Evaluating the effect of visual grounding on the language stream

We first extract the word embeddings from the language model, which always has a Bert-base structure but with different levels of visual grounding (**Bert**: no visual grounding; **Grounded**: visual grounding of natural language; **Relational Grounded**: visual grounding of object relations). To do this, we input every single word (or phrase) preceded with a special token [CLS] and followed by a special token [SEP] (according to the original Bert [7] paper) into the language stream, and use the average pooled output from the last hidden layer as the extracted word embedding. Since a few output feature channels have much larger standard deviations than other feature channels, we further use the mean and standard deviation of the output representation of the $30,522$-token vocabulary (which defines the embedding layer in Bert [8]) to standardize each single word representation. The same process is applied to both the Bert model and the visually grounded language models. For each input word (or phrase), its output embedding is a $d$-dimensional vector ($d = 768$). All embeddings of commonly used English words from the vocabulary set $S$ (defined by the SemCat dataset [9]) form a set of vector representations in this high-dimensional semantic space.

## B.1 Extended results on principal component analysis

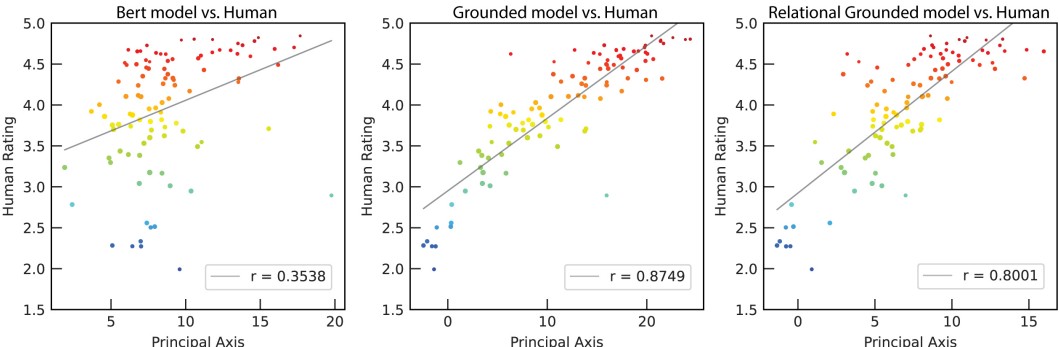

Figure 7: The first principal component in word representation space captures concrete-abstract axis only after visual grounding. The color-coding of each dot indicates the averaged human-rated concreteness score of a category (blue: abstract; red: concrete).

Table 6: Correlation between the $1^{\text{st}}$ principal axis and human-rated word concreteness

| Group | Correlation (Pearson's r) | | |
|---|---|---|---|
| | Bert | Grounded | Relational Grounded |
| word-level | 0.1040 | 0.6615 | **0.6948** |
| category-level | 0.3538 | **0.8749** | 0.8001 |

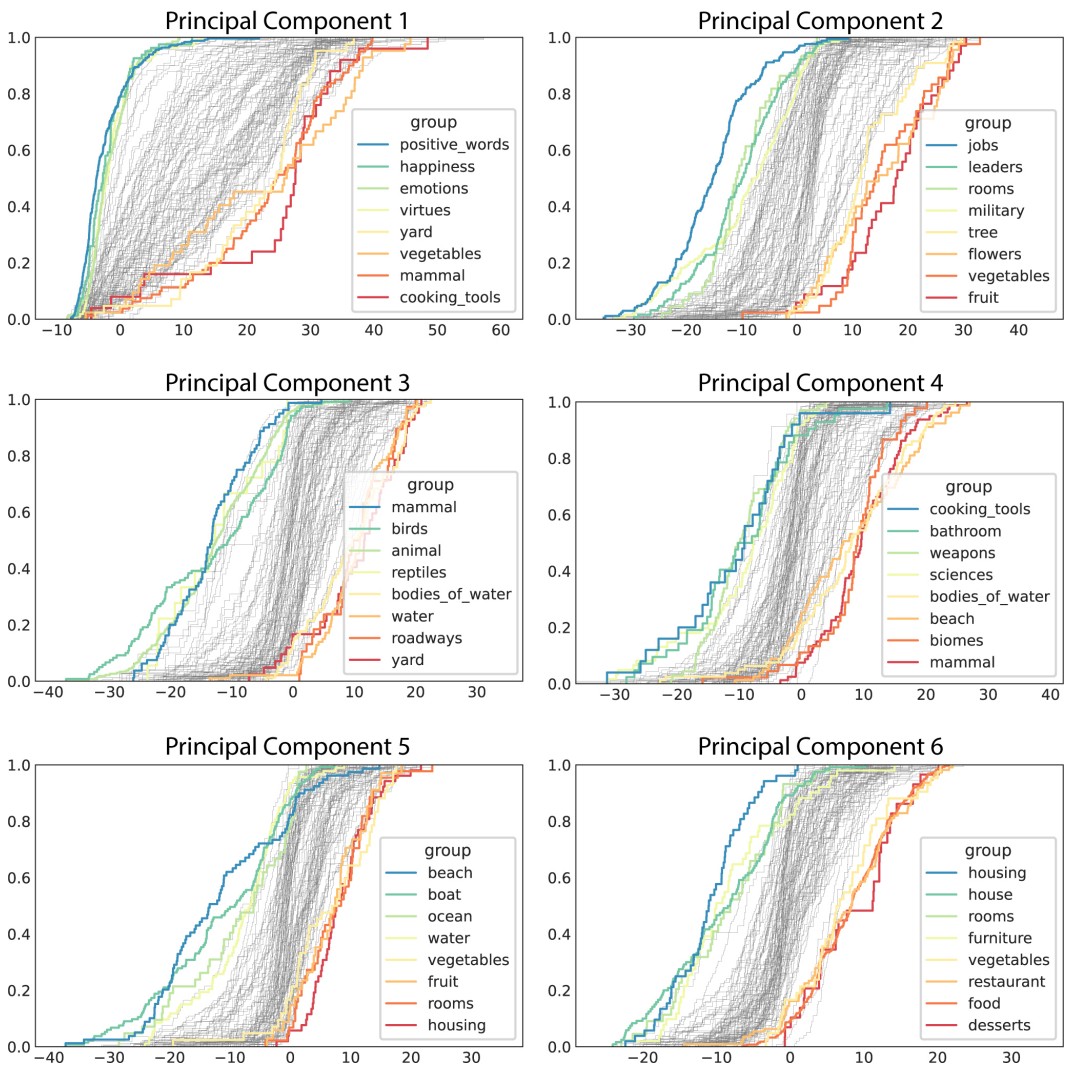

Figure 8: Other principal components in the visually grounded word representation space. Each plot shows a set of cumulative distribution functions (CDFs) for different word categories after projecting to a principal component. The resulting principal dimensions capture the semantic attributes that can be interpreted by human intuition. PC1: abstract vs. concrete; PC2: human vs. non-human; PC3: object vs. scene; PC4: artificial vs. natural; PC5: outdoor vs. indoor; PC6: non-food vs. food.

### B.1.1  2D visualization of the first three principal axes

To facilitate the understanding of how conceptual representations are organized in the grounded semantic space, we visualize the 100 word categories in a linear subspace spanned by the first three principal axes. We first color-code each word category by using an RGB code: (PC1, red), (PC2, green), (PC3, blue). Only for this color-coding purpose, the coefficients of each principal component are linearly re-scaled into the range $[0, 1]$. We then project the three dimensional representations of the 100 word categories further into three 2D planes, as shown in Fig. 9 (PC2 vs. PC3), Fig. 10 (PC1 vs. PC2), and Fig. 11 (PC1 vs. PC3).

The result of PC 2 vs. PC 3 (Fig. 9) suggests that the first quadrant represents concepts describing natural scenes (e.g. biomes, rocks), the second quadrant represents concepts related to scenes with human activities (e.g. roadways, rooms), the third quadrant encodes human-related non-scene concepts (e.g. jobs, musical instruments), the fourth quadrant encodes non-human objects (e.g. animal, foodweb). The abstract words (e.g. emotions, happiness) are squeezed around the origin in this projected representation.

In the 2D projection of PC1 vs. PC2 (Fig. 10) or PC1 and PC3 (Fig. 11), we also observe that although concrete concepts are distributed and scattered widely in the semantic space, the abstract concepts tend to be squeezed around the origin. This is perhaps caused by the lack of information supporting rich representations for emotional words since we only grounded the language model in vision.

**Explainable Pricipal Components in the Visually Grounded Semantic Space**

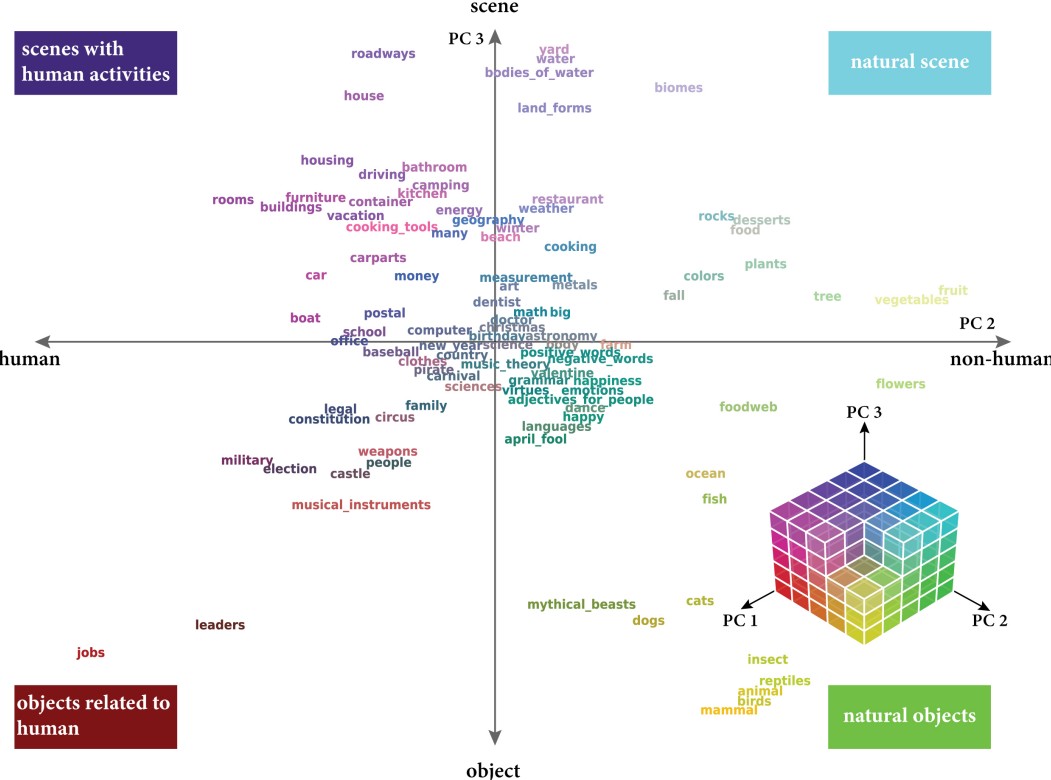

Figure 9: 2D visualization of PC2 and PC3.

## Explainable Pricipal Components in the Visually Grounded Semantic Space

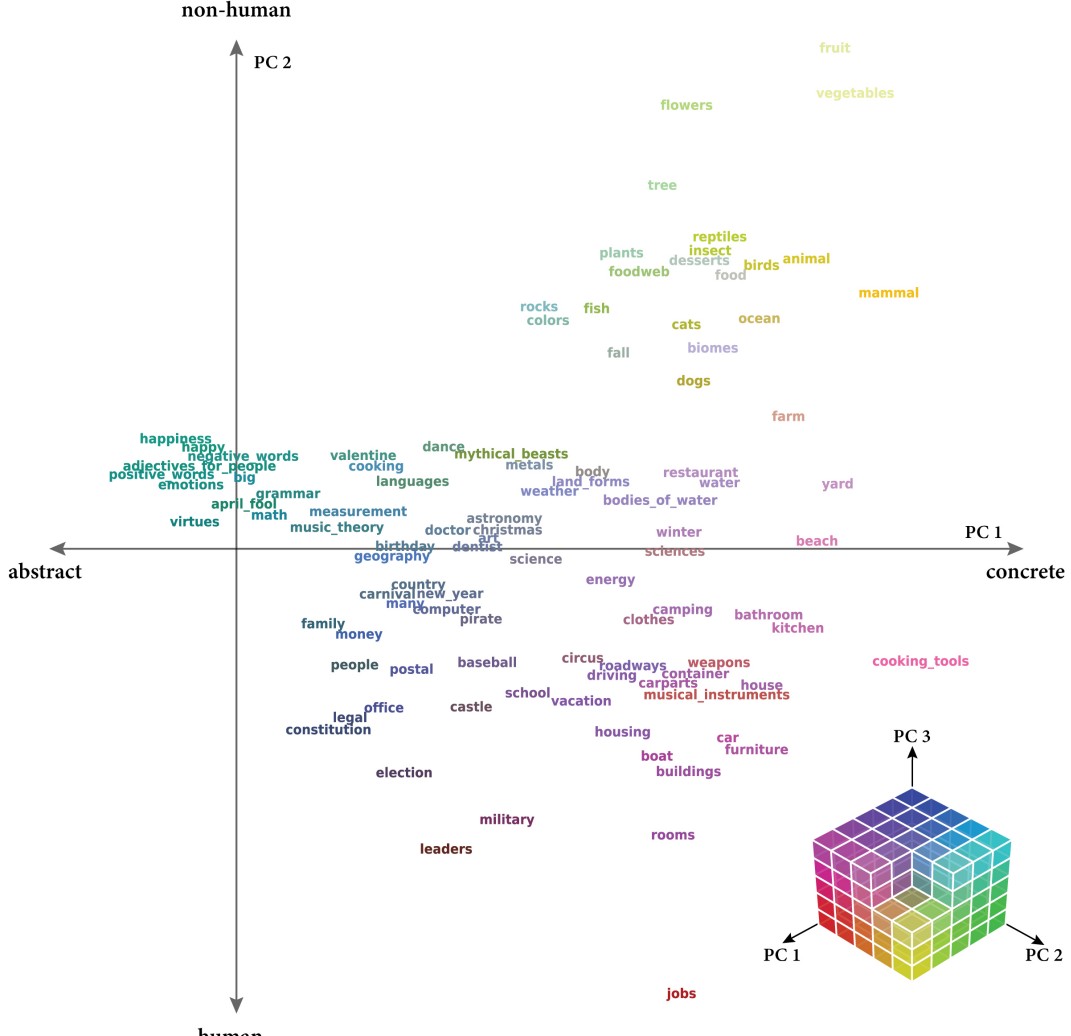

Figure 10: 2D visualization of PC1 and PC2.

**Explainable Pricipal Components in the Visually Grounded Semantic Space**

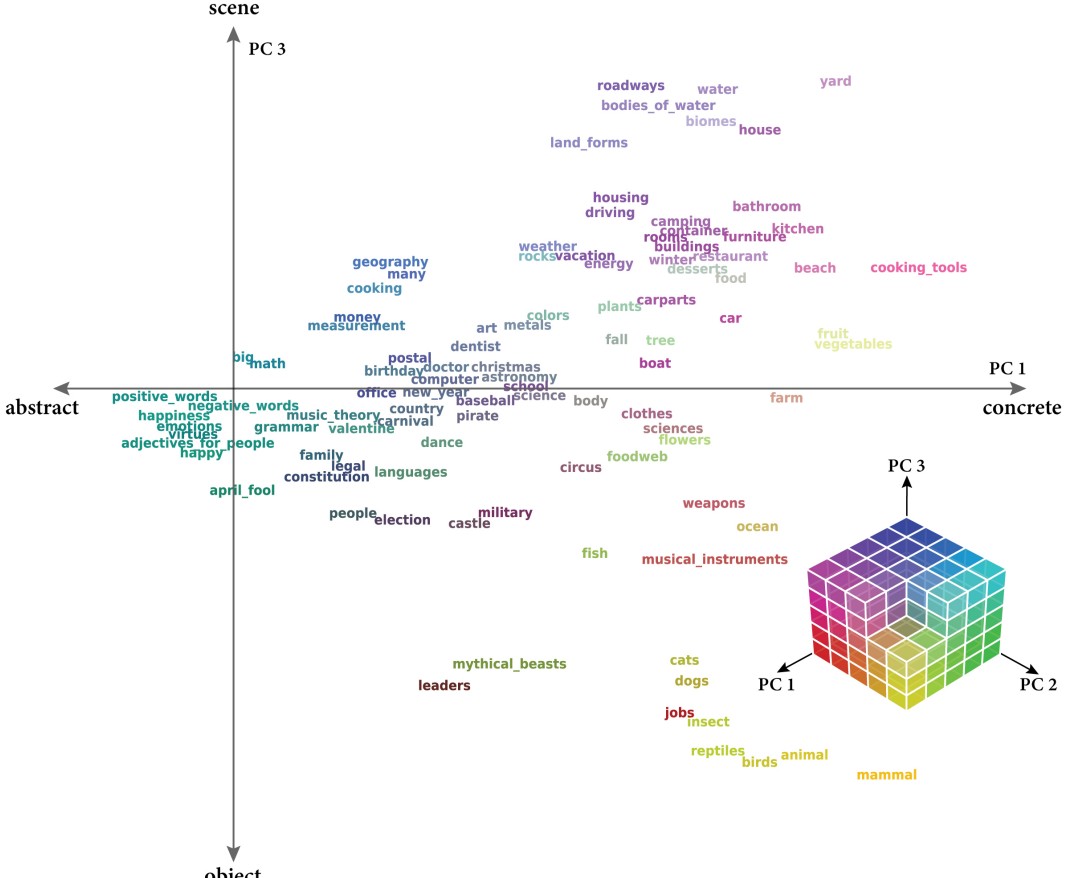

Figure 11: 2D visualization of PC1 and PC3.

## B.2 Extended results on semantic norm prediction

To investigate whether the visually grounded word embeddings capture semantic norms defined by humans, we train a logistic regression model with L1 regularization from the word embeddings to predict each binary semantic feature defined in the CSLB dataset [10].

Since many binary semantic norms in the CSLB dataset contain very few positive word samples, we first filter out the ones with fewer than 5 positive samples, which results in 390 out of 2725 feature norms, dividing into five feature types: 156 "visual perceptual" features (e.g. `has_wheels`); 29 "other perceptual" features (e.g. `has_flavors`); 94 "functional" features (e.g. `does_cut`); 65 "encyclopaedic" features (e.g. `is_dangerous`); 46 "taxonomic" features (e.g. `is_clothing`). For the $i$-th semantic norm, we build a binary classifier with a logistic regression model $p^i$ ([11]):

$$p^i(y_{ij} = 1|\boldsymbol{x_j}) = \sigma(\boldsymbol{w_i}^T \boldsymbol{x_j}), \tag{7}$$

Here $\boldsymbol{x_j}$ is the word representation of the $j$-th word $x_j$ after projecting onto principal axes. $\boldsymbol{w_i}$ is a linear weight specific to the $i$-th semantic norm. $y_{ij} \in \{0, 1\}$ is the binary label indicating whether the $x_j$ holds the $i$-th semantic norm. To avoid over-fitting, we add an L1-norm as a sparsity constraint.

$$\boldsymbol{w_i}^* = \underset{\boldsymbol{w_i}}{\arg\min} \left( -\sum_j \left[ y_{ij} \log\left(\sigma(\boldsymbol{w_i}^T \boldsymbol{x_j})\right) + (1 - y_{ij}) \log\left(1 - \sigma(\boldsymbol{w_i}^T \boldsymbol{x_j})\right) \right] + \lambda_i \|\boldsymbol{w_i}\|_1 \right) \tag{8}$$

The regularization parameter $\lambda_i$ is determined by a leave-one-out cross-validation to minimize the following objective function:

$$\mathcal{L}_i(\lambda_i) = \sum_j \mathcal{L}_{ij}(\lambda_i) \tag{9}$$

Suppose $P_i = \{k|y_{ik} = 1\}$ and $N_i = \{k|y_{ik} = 0\}$ are the sets consisting of positive and negative word samples for the $i$-th semantic feature, respectively ($|P_i \cup N_i| = 638$, i.e., the total number of words in the CSLB dataset).

$$\mathcal{L}_{ij}(\lambda_i) = \frac{1}{|P_i|} \sum_{k \in P_i, k \neq j} (\log p^i_{\lambda_i,j}(y_{ik} = 1|\boldsymbol{x_k})) + \frac{1}{|N_i|} \sum_{k \in N_i, k \neq j} (\log p^i_{\lambda_i,j}(y_{ik} = 0|\boldsymbol{x_k})) \tag{10}$$

$j$ indicates the left-out word sample, and $p^i_{\lambda_i,j}$ is the trained regression model with regularization parameter $\lambda_i$. After $\lambda_i$ is determined, we train the L1-normed logistic regression model $p^i$ with all word samples and calculate the F1-score:

$$\text{F1} = \frac{\texttt{tp}}{\texttt{tp} + \frac{1}{2}(\texttt{fp} + \texttt{fn})} \tag{11}$$

where `tp` refers the number of true positive cases, `fp` refers the number of false positive cases, `fn` refers the number of false negative cases. We then pairwisely compare the F1-score for each semantic norm across language models with different levels of visual grounding and test the statistical significance with a one-sided Wilcoxon Signed Rank Test (as shown in the main Fig. 4). Since we add a strong regularization term to avoid over-fitting, the results suggest 230 out of 390 semantic norms are not predictable by the ungrounded Bert model, while only 143 and 129 semantic norms are not predictable by the Grounded model and the Relational Grounded model respectively.

To better understand the details of this dataset and the corresponding results, we listed the top-5 semantic norms that became better predictable (according to the F1-score) after visual grounding in Table 7.

Table 7: Top-5 semantic norms that are better predictable after visual grounding.

| Feature type | Grounded model | Relational Grounded model |
|---|---|---|
| visual perceptual | `has_wheels,`
`has_a_handle_handles,`
`has_skin_peel,`
`has_pages, has_a_back` | `has_a_picture_pictures,`
`has_pages, has_a_barrel,`
`has_skin_peel,`
`has_a_seat_seats` |
| other perceptual | `is_heavy, is_warm,`
`does_smell_good_nice,`
`is_juicy, has_flavours` | `is_warm, is_heavy,`
`has_flavours, is_juicy,`
`does_smell_good_nice` |
| functional | `does_fly,`
`does_contain_hold,`
`does_store, does_heat,`
`is_used_to_see` | `does_fly, does_heat,`
`does_cut,`
`is_used_in_cooking,`
`does_contain_hold` |
| encyclopaedic | `is_dangerous,`
`is_found_in_seas,`
`has_information,`
`is_healthy,`
`does_grow_on_trees` | `is_dangerous,`
`has_information,`
`does_grow_on_trees,`
`is_found_in_seas,`
`is_found_in_kitchens` |
| taxonomic | `is_clothing,`
`is_a_weapon,`
`is_a_vehicle,`
`is_a_vegetable,`
`is_transport` | `is_clothing,`
`is_a_vehicle,`
`is_a_vegetable,`
`is_medicine,`
`is_a_container` |

### B.3 Extended results on word categorization

#### B.3.1 The modified version of Sihlouette coefficient

Suppose $N$ is the number of word categories, $\boldsymbol{x_i}$ is the embedding of word $i$ which belongs to the category $C_i$, let

$$a(i) = \frac{1}{|C_i| - 1} \sum_{j \in C_i, j \neq i} d(\boldsymbol{x_i}, \boldsymbol{x_j}), \tag{12}$$

$$b(i) = \frac{1}{N - 1} \sum_{k \neq i} \frac{1}{|C_k|} \sum_{j \in C_k} d(\boldsymbol{x_i}, \boldsymbol{x_j}), \tag{13}$$

$$s(i) = \frac{b(i) - a(i)}{\max\left(a(i), b(i)\right)} \tag{14}$$

where the distance metric $d$ between word embeddings is measured as the cosine distance: $d(\boldsymbol{x_i}, \boldsymbol{x_j}) = 1 - \cos(\boldsymbol{x_i}, \boldsymbol{x_j})$. Here we used average instead of minimum when calculating the denominator in $b(i)$ (Eq. 13) since these 100 word categories are not mutually exclusive (e.g. `mammal`, `bird`, `fish` are overlapped with `animal` in SemCat dataset).

#### B.3.2 Word categorization performance compared across different training settings

We also compare the word categorization performance after the visual grounding of natural language with different training settings (Fig. 12). The results suggest that earlier grounding (i.e. more learnable layers in Bert) tends to show better clustering performance on human-defined word categories. The models with frozen query and key transformations in Bert self-attention layers (blue bars) show similar categorization performance as the ones with learnable query and key weights for cross-modal training (orange bars). Models trained with a larger dropout rate (0.3; as shown in opaque bars) have significantly higher performance on word categorization than its counterpart with a smaller dropout rate (0.1; as shown in transparent bars).

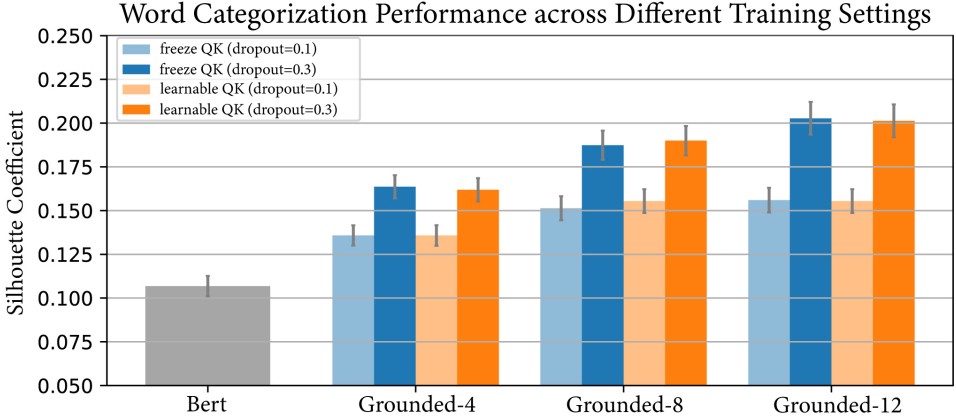

Figure 12: Word categorization performance compared across different training hyperparameters.

#### B.3.3 Word categorization performance is uncorrelated with its occurrence rate

In order to validate whether a better clustering performance on word $w_i$ results from a higher sampling rate in the training dataset, we further calculate the correlation between the Silhouette coefficient $s(i)$ and the training occurrence rate of word $w_i$ for all words in the SemCat dataset. The result (Fig. 13) rejects this hypothesis by showing a weak correlation value between these two terms for both category-level ($r = -0.28$) and word-level ($r = -0.07$) analysis.

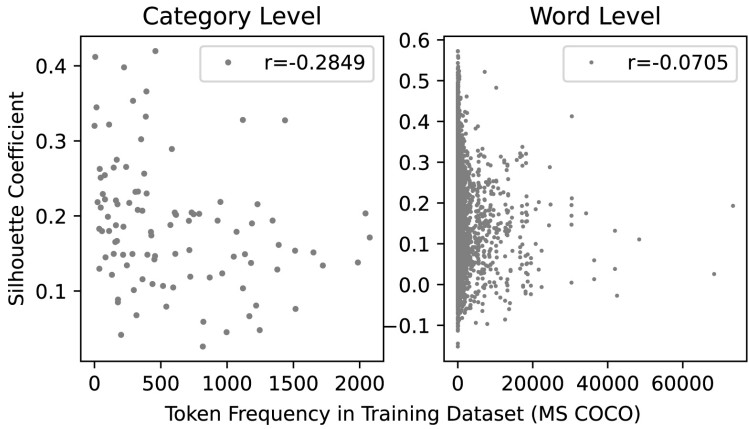

Figure 13: Word categorization performance is uncorrelated with its occurrence rate during training.

### B.3.4 More examples on representations of word subcategories

The following results are examples of **vehicle** (Table 8, Fig 14), **animal** (Table 9, Fig 15), **food** (Table 10, Fig 16), and **room** (Table 11, Fig 17) subcategories. For each subcategory, we first pick up three query words as the prototype for each subcategory (e.g. boat, car, airplane for **vehicle**). Then for each language model, we use the cosine similarity to sort out the top-15 closest words to each of these query words, as shown in the columns of these tables. We observe that the top similar words are well-aligned with the subcategory defined by the query word only after visual grounding. For example, all words in the column under "Grounded"/"Relational Grounded" for the boat query belong to water transportation, but words *skeleton, nation, men, bike* found by Bert model are not water transportation. We further visualize the representation of the words in each subcategory from the "Grounded" model, by first calculating its cosine similarity to each of the query word (resulting in a three-dimensional vector), and then projecting these three-dimensional vectors into the 2D plane spanned by a pair of query word. The visualization results shown in the following figures further demonstrate that the subcategories are separable only after visual grounding.

Table 8: Top-15 words for **vehicle** subcategories

| query | Bert | Grounded | Relational Grounded |
|---|---|---|---|
| boat | canoe, sailboat, submarine, skeleton, nation, sailing, men, bike, ballast, boating, raft, motorboat, paddle, yacht, kayak | tugboat, yacht, riverboat, gunboat, canoe, boating, barge,dinghy, raft, sailboat, ship, steamboat, steamer, trawler, watercraft | riverboat, gunboat, canoe, sailboat, dinghy, tugboat, yacht, motorboat, ship, steamer, barge, steamboat, submarine, ferry, steamship |
| car | vehicle, jeep, sedan, truck, jaguar, auto, driver, motorcycle, chassis, motor, bike, boat, horse, speeding, automobile | sedan, suv, vehicle, automobile, limo, jeep, limousine, taxi, drive, traffic, auto, pickup, van, roadster, truck | sedan, limo, automobile, suv, jeep, van, limousine, taxi, truck, buggy, vehicle, cart, motorcycle, auto, convertible |
| airplane | plane, aircraft, propeller, automobile, airport, bird, flight, parrot, turbulence, fly, kite, butterfly, rocket, motorcycle, takeoff | plane, jet, flight, aircraft, takeoff, airport, corsair, pilot, hangar, undercarriage, missile, propeller, nuclear, nautical, flyby | plane, jet, aircraft, corsair, flight, balloon, missile, takeoff, automobile, cockpit, hangar, avian, rocket, freighter, nuclear |

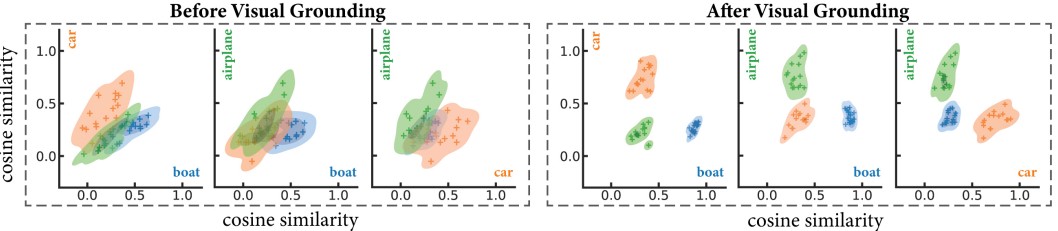

Figure 14: The distribution of representational similarity on **vehicle** words.

Table 9: Top-15 words for **animal** subcategories

| query | Bert | Grounded | Relational Grounded |
|---|---|---|---|
| dog | pig, animal, bike, horse, mule, donkey, squirrel, bicycle, goat, monkey, motorcycle, moose, cat, gorilla, mouse | puppy, doge, terrier, canine, pug, hound, beagle, bulldog, dogwood, pup, mutt, spaniel, chihuahua, retriever, shepherd | puppy, doge, terrier, pug, canine, bulldog, beagle, hound, mutt, greyhound, bobcat, spaniel, hag, tomcat, donkey |
| goose | scare, geese, cow, calf, neighbor, puddle, flu, battleship, hog, displeasure, plank, herring, stir, scrambled, sock | geese, eagle, rooster, gull, pigeon, owl, duck, crow, parrot, partridge, falcon, harrier, sparrow, vulture, warbler | geese, eagle, pigeon, owl, parrot, duck, sparrow, rooster, falcon, crow, seagull, gull, warbler, partridge, harrier |
| horse | stallion, mule, dog, bike, trainer, boat, mare, car, animal, men, motorcycle, human, mountain, athlete, chestnut | stallion, mule, mare, donkey, seahorse, ox, steer, bull, unicorn, chestnut, foal, camel, oxbow, antelope, cow | stallion, mule, mare, seahorse, donkey, camel, bull, cow, cattle, ox, bison, deer, lassie, dog, animal |

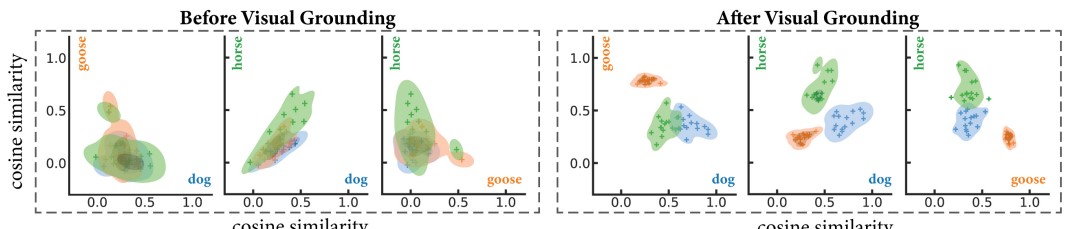

Figure 15: The distribution of representational similarity on **animal** words.

Table 10: Top-15 words for **food** subcategories

| query | Bert | Grounded | Relational Grounded |
|---|---|---|---|
| drink | eat, feed, swallow, spend, study, spill, breathe, treat, bite, cough, drop, give, relax, wash, bleed | beverage, soda, cola, coke, juice, rum, bottle, champagne, drunk, flask, blender, mug, cup, blend, coffee | beverage, soda, cola, coke, juice, rum, flask, bottle, lemonade, coffee, cup, blend, mug, blender, brew |
| fruit | flower, foliage, citrus, flowers, plant, orchid, shrub, poisonous, inflorescence, eggs, omnivorous, seedling, nut, pineapple, snail | citrus, grape, strawberry, pear, pineapple, lemon, grapefruit, peach, mango, nuts, seeds, tomato, beets, tangerine, apple | citrus, strawberry, pineapple, grape, grapefruit, lemon, mango, peach, pear, apple, ripe, nuts, seeds, cherry, cranberry |
| vegetable | potato, beans, vegetables, cheese, mustard, beef, boiled, chicken, tomato, milk, corn, pig, bread, grape, grains | vegetables, salad, greens, botany, plantain, weeds, crops, algae, herb, legumes, perennial, lettuce, sprouts, vegetation, sprout | vegetables, botany, greens, legumes, salad, herb, plantain, herbs, lettuce, celery, crops, pomegranate, algae, asparagus, carrot |

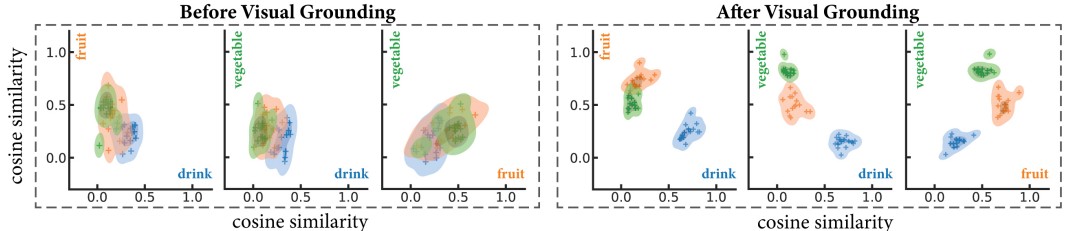

Figure 16: The distribution of representational similarity on **food** words.

Table 11: Top-15 words for **room** subcategories

| query | Bert | Grounded | Relational Grounded |
|---|---|---|---|
| bathroom | restroom, bedroom, bath, kitchen, toilet, laundry, refrigerator, couch, bathtub, dresser, shower, hallway, mirror, towel, backyard | restroom, bath, shower, bathtub, toilet, sink, vanity, wash, tub, mirror, towel, soap, hygiene, hallway, shave | restroom, kitchen, cafeteria, bedroom, bath, room, shower, hospital, office, gym, classroom, pantry, hotel, gymnasium, motel |
| bedroom | bathroom, bed, room, dresser, downstairs, apartment, kitchen, condo, couch, upstairs, backyard, mattress, hallway, bath, attic | bed, mattress, pillow, room, closet, condominium, crib, dresser, apartment, motel, upstairs, cot, dorm, blanket, robe | room, closet, dorm, hotel, kitchen, apartment, motel, bathroom, household, dormitory, office, hostel, classroom, cafeteria, house |
| kitchen | refrigerator, bathroom, couch, fireplace, backyard, laundry, barn, furniture, sofa, basement, toilet, bedroom, cupboard, stairs, driveway | pantry, counter, cupboard, household, galley, stove, cook, microwave, oven, refrigerator, freezer, kettle, furnace, washer, chef | pantry, cafeteria, household, bathroom, bedroom, restroom, room, office, showroom, classroom, restaurant, garage, parlor, gym, dugout |

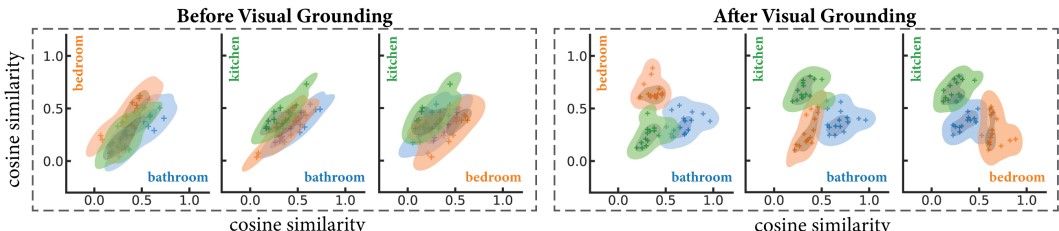

Figure 17: The distribution of representational similarity on **room** words.

## B.4 Extended results on vision based concept composition

For evaluating the vision-based concept composition described in main text Section 4.4, we also visualize the ranking change of words that are most similar to a query phrase before and after visual grounding with a slope chart. In addition to the `striped horse` example shown in the main text, we also plot the slope chart for example query `red fruit` (Fig. 18). The result suggests both models found most similar words belonging to fruit/plant, but after grounding the color information learned from the visual stream enhance the semantic representation of words like *tomato, strawberry, cranberry* to be closer to `red fruit`.

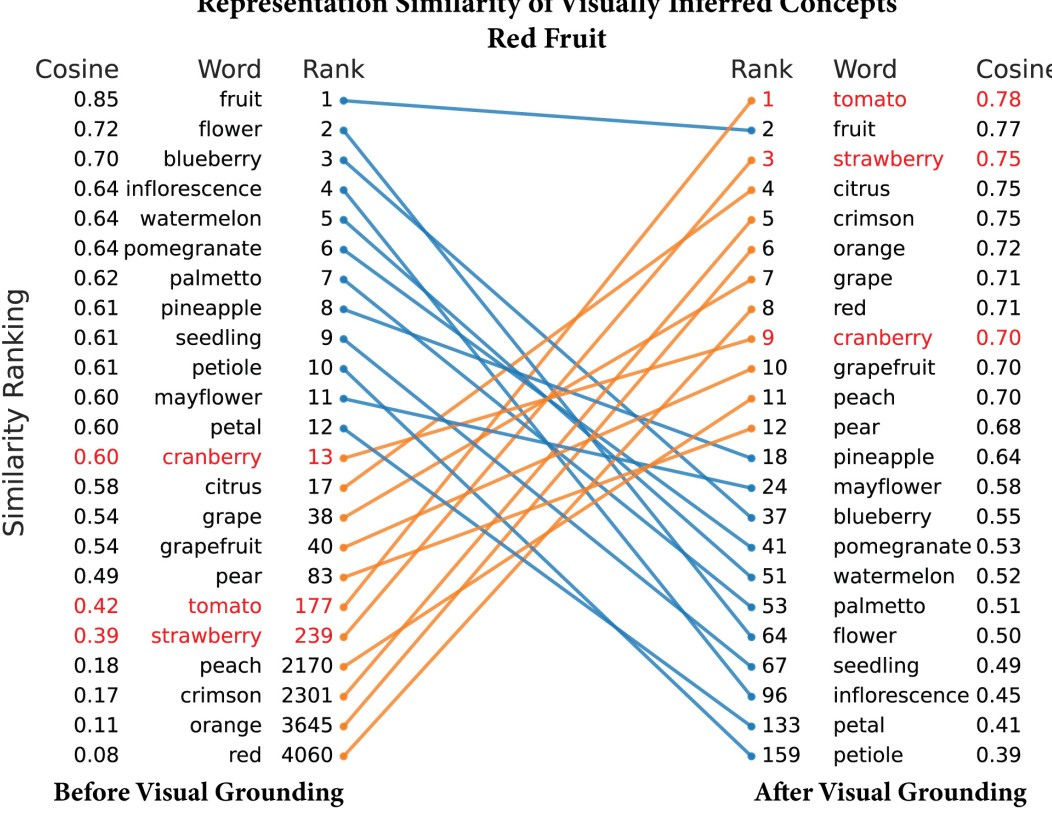

Figure 18: Concept composition based on visual knowledge (**red fruit**).

## B.5   Supplementary method for multimodal image search

To implement multimodal image search, we first add two additional heads ($F_V$ and $F_L$) to the image key and langauge query (after being concatenated across all attention heads along the feature dimension) from the cross-modal attention module in Section 3.2.

$$\boldsymbol{Q}_I = F_V(\text{Key}_V) = \frac{1}{HW} \sum_{i,j} \left( \left( \text{ReLU}(\text{Key}_V[i,j,:]\boldsymbol{W}_V^1 + \boldsymbol{b}_V^1) \right) \boldsymbol{W}_V^2 + \boldsymbol{b}_V^2 \right), \qquad (15)$$

$$\boldsymbol{Q}_W = F_L(\text{Query}_L) = \frac{1}{K} \sum_{k} \left( \left( \text{ReLU}(\text{Query}_L[k,:]\boldsymbol{W}_L^1 + \boldsymbol{b}_L^1) \right) \boldsymbol{W}_L^2 + \boldsymbol{b}_L^2 \right), \qquad (16)$$

where $\boldsymbol{W}$s and $\boldsymbol{b}$s are the weights and biases of the linear transformations in these two head functions, with size $d \times d$ and $d \times 1$ ($d = 768$). $H$ and $W$ are the height and width of the image feature output ($H = W = 14$). $K$ is the number of words in an image caption, which varies for different language inputs. $F_V$ and $F_L$ are applied to visual and textual representations ($\text{Key}_V$ or $\text{Query}_L$) in the joint space respectively. After average pooling the outputs, we get a single vector representation for either an image (denoted as $\boldsymbol{Q}_I \in \mathbb{R}^d$) or a text (denoted as $\boldsymbol{Q}_W \in \mathbb{R}^d$).

These two representations are L2 normalized (Eq. 17) and aligned into a shared space, by freezing the pretrained two-stream model and only finetuning the transformation heads $F_V$ and $F_L$ with contrastive loss to match paired images and texts in terms of their cosine similarity using the MS COCO dataset.

$$\boldsymbol{Q}_I \leftarrow \frac{\boldsymbol{Q}_I}{\|\boldsymbol{Q}_I\|_2}, \quad \boldsymbol{Q}_W \leftarrow \frac{\boldsymbol{Q}_W}{\|\boldsymbol{Q}_W\|_2}, \qquad (17)$$

$$\boldsymbol{Q}_{\text{search}} = (1-\alpha)\boldsymbol{Q}_I + \alpha\boldsymbol{Q}_W. \qquad (18)$$

Then to construct a multimodal search query $\boldsymbol{Q}_{\text{search}}$, we use a linear combination of a pair of image query $\boldsymbol{Q}_I$ and text query $\boldsymbol{Q}_W$ (since they are now in the same representational space) weighted by a scalar $\alpha$ (Eq. 18). When $\alpha = 0$, only the image is used as a query for search and retrieval. When $\alpha = 1$, only the word is used as the search query. And when $\alpha = 0.5$, the image and text inputs contribute equally to the search query. The weight $\alpha$ can vary continuously in the range from 0 to 1. If the visual and text-informed semantics share a common semantic space, then varying alpha is expected to result in the retrieved images to vary their contents according to progressive and intuitive transition from the image to the text.

### B.6 Evaluate the grounded language model on GLUE benchmark

Similar to the evaluation in [7], we further test the language model before and after visual grounding on the General Language Understanding Evaluation (GLUE) benchmark [12]. For this purpose, the Bert encoder in the language models is fixed, while only a pooling layer (shared across all tasks in GLUE) and a linear classification layer (specific for each task in GLUE) are trainable. We use the training and evaluation codes from the `jiant` [1] package. The testing results are submitted to and evaluated by the GLUE benchmark website [2].

Table 12: Model performance on GLUE benchmark.

| Model | CoLA | SST-2 | MRPC | STS-B | QQP | MNLI-(m/mm) | QNLI | RTE | Average |
|-------|------|-------|------|-------|-----|-------------|------|-----|---------|
| Bert | 40.7 | 92.6 | 87.8 | 81.8 | 71 | 83.3/82 | 89.4 | 73.8 | 78.04 |
| Grounded-4 | 37.1 | 91.4 | 86 | 83.3 | 70.8 | 82.7/81.5 | 88.9 | 73 | 77.19 |
| Grounded-8 | 38.6 | 91.5 | 86.3 | 83.4 | 70.8 | 82/81 | 87.9 | 72.2 | 77.08 |
| Grounded-12 | 37.2 | 92.6 | 86.5 | 82.3 | 70.5 | 82.3/81.7 | 89.2 | 72 | 77.10 |
| Relational-2 | 38 | 92.8 | 84.6 | 81.8 | 70.4 | 83.1/81.8 | 89.2 | 71.7 | 77.04 |

Table 12 summarizes the results. `Grounded-`$k$ models have $k$ learnable layers in Bert for visual grounding of natural language with MS COCO dataset. `Relational-2` model is trained from `Grounded-`8 by finetuning 2 Bert layers for visual grounding of object relations.

The results suggest that in general the language model has a slightly decreased performance on GLUE after visual grounding, although the models tested here all share the same architecture as Bert. Allowing more learnable parameters during visual grounding tends to result in worse performance for natural language understanding. This is unsurprising since the grounding process is based on matching short captions (MS COCO dataset) or phrases (Visual Genome dataset) to visual contents, which may not require extensive capacity for textual processing. Future study is needed to reconcile the trade-off performance between visual grounding and natural language understanding.

## C  Code

We release the code for training and testing the proposed models at https://github.com/yizhen-zhang/VG-Bert.

---

[1]Jiant package: https://github.com/nyu-mll/jiant/
[2]https://gluebenchmark.com/