# OpenReview forum: "Explainable Semantic Space by Grounding Language to Vision with Cross-Modal Contrastive Learning"
_NeurIPS.cc/2021/Conference — NeurIPS 2021 Poster_

### Official Review · Reviewer_zUxZ · 2021-06-29

**Rating:** 7
**Confidence:** 4

**Summary:**

The paper implements a model which learns word representations from
text grounded in images. The model is composed of a BERT-based text
stream, a VGG-based image stream, and visual relation classification
module. The text and image components are pre-trained and fine tuned
on the COCO dataset of captioned images as well as on Visual Genome
relations. The main contribution of the paper is the large number of
analyses carried out on the induced grounded word embeddings, compared
to plain BERT embedddings. The paper shows some evidence that the
principal components of the grounded embedding space correspond to
interpretable semantic dimensions (i.e. concrete-abstract), that
words' semantic features can be predicted based on embeddings. The
paper also claims that the model is capable of compositional
understanding.


**Limitations And Societal Impact:**

Limitations are not explicitly discussed. The preliminary nature and
small scale of some of the experiments (e.g. section 4.4) should be
acknowledged. The word "explainable" suggests that the architecture is
designed to be explainable, but in fact it doesn't seem to be the case: rather,
analyses are carried out which provide some insight into the learned
representations.



**Main Review:**

## Strengths

The main idea of visual grounding  has been of interest in
computational linguistics for a long time, and has been investigated
computationally many times. This paper main contributions are the use
of state-of-the-art neural architectures and pretrained models, and an
in-depth study of the nature of the learned embeddings. The results
range from compelling (the interpretable principal components) to
suggestive but preliminary (compositionality), but overall show
convincingly that in this particular incarnation of grounding does
make a major contribution to make the word representations more
human-like.

## Weaknesses

The discussion of and comparison to earlier related work is very
incomplete. Given the analytical focus of the paper quantitative
comparisons to alternative models are not absolutely necessary but it
would be good to at least see a more in-depth related work
section. There are a lot of papers that could be mentioned here;
a small sample follows:


- Bordes, P., Zablocki, E., Soulier, L., & Piwowarski, B. (2019,
  November). Incorporating Visual Semantics into Sentence
  Representations within a Grounded Space. In Proceedings of the 2019
  Conference on Empirical Methods in Natural Language Processing and
  the 9th International Joint Conference on Natural Language
  Processing (EMNLP-IJCNLP) (pp. 696-707).

- Kiros, J., Chan, W., & Hinton, G. (2018, July). Illustrative
  language understanding: Large-scale visual grounding with image
  search. In Proceedings of the 56th Annual Meeting of the Association
  for Computational Linguistics (Volume 1: Long Papers) (pp. 922-933).

- Chrupała, G., Kádár, Á., & Alishahi, A. (2015, July). Learning
  language through pictures. In Proceedings of the 53rd Annual Meeting
  of the Association for Computational Linguistics and the 7th
  International Joint Conference on Natural Language Processing
  (Volume 2: Short Papers) (pp. 112-118).


**Time Spent Reviewing:**

3

---

> ### Author Response · Authors · 2021-08-10
> **Response to Reviewer zUxZ**
>
> We agree with the reviewer’s summary and appreciate the reviewer’s effort and comments. Overall, our work leverages the concept of “grounding language in vision” with the state-of-the-art neural network architecture. Although neither the conceptual nor technical framework is entirely new by itself, their combination remains relatively less explored.
>
> Unique contributions from our work are: 1) visual grounding is applied to not only concepts but also relationships between concepts, and 2) the language model after visual grounding gives rise to a continuous and interpretable semantic space as demonstrated in a comprehensive set of experiments. We believe our work makes a solid and timely contribution to grounded language learning. See our detailed replies to specific comments.
>
> Grounding language learning in vision is not a new idea. It has been of increasing interest in the fields of computational linguistics and machine learning. According to the reviewer’s suggestion, we will extend our discussion about related prior work, including those papers suggested by the reviewer, tentatively as below.
>
> - Grounding language in vision has been of increasing interest in computational linguistics and machine learning. A common strategy is to fuse words with related visual information in terms of perceptual norms [3], bag-of-visual-word [4,5], or learnable features [6,7,8,10]. The models used for vision-language fusion evolve alongside those for natural language processing, such as Latent Dirichlet Allocation (LDA) [4], log-bilinear model (like GloVe) [11], Skip-gram model (like word2vec) [6,9], recurrent neural network [7]. Findings from prior work suggest that visual grounding may refine the distribution and interpretability of language representations [4,5,7,8,9,10,11,12] and facilitate cross-modal tasks [7,8,10,11,12].
>
> Although we are inspired by the prior work, our approach is different in two aspects. First, we use the transformer [13,14] as the backbone for the language model with or without visual grounding. Second, we use contrastive learning [15,16] for visual grounding of both words and word relations. In particular, visual grounding of pairwise word relations has not been explored, to the best of our knowledge.
>
> ---
>
> Furthermore, although both our and prior work evaluates grounded word embeddings with similar metrics, such as word concreteness, feature norm prediction, word clustering, our results highlight a novel finding that the visually grounded semantic space exhibits principal dimensions interpretable by human intuitions and neurobiological knowledge. In particular, the first principal dimension is highly correlated (r=0.87) with human-rated concreteness. As the reviewer agrees, this finding is compelling with potentially profound implications to both computational linguistics, cognitive science, and interpretable machine learning.
>
> According to the reviewer’s comments, we will revise the discussion section to explicitly discuss the limitations of our work, tentatively as below.
>
> - Several limitations of our work are noteworthy. The datasets used to train our model are orders of magnitude smaller than those used in recent studies [15,16]. Scaling up the model training with increasingly larger datasets is expected to greatly improve the model’s performance for cross-modal tasks, while generally preserving the interpretability of the grounded language representations as described herein. Some of our experiments and results are preliminary for illustrative purposes and await more comprehensive and quantitative evaluation in future studies, especially with more downstream vision-language tasks. Whereas our evaluation focuses on the language model, grounding language to vision may also have refined the visual stream, awaiting further evaluation against visual tasks, as demonstrated elsewhere [15,16].
>
> ---
>
> Lastly, we agree with the reviewer that the word “explainable” in the title of this paper refers to the interpretability of the grounded semantics, instead of the model architecture. If the paper is accepted by NeurIPS, we will try to replace the word “explainable” by “interpretable”, if we are allowed to do that.
>
> ---
>
> References:
>
> [1] Radford, A., Kim, J. W., Hallacy, C., Ramesh, A., Goh, G., Agarwal, S., ... & Sutskever, I. (2021). Learning transferable visual models from natural language supervision. arXiv preprint arXiv:2103.00020.
>
> [2] Jia, C., Yang, Y., Xia, Y., Chen, Y. T., Parekh, Z., Pham, H., ... & Duerig, T. (2021). Scaling up visual and vision-language representation learning with noisy text supervision. arXiv preprint arXiv:2102.05918.
>
> [3] Silberer, C., & Lapata, M. (2012, July). Grounded models of semantic representation. In Proceedings of the 2012 Joint Conference on Empirical Methods in Natural Language Processing and Computational Natural Language Learning (pp. 1423-1433).
>
> [4] Bruni, E., Tran, N. K., & Baroni, M. (2014). Multimodal distributional semantics. Journal of artificial intelligence research, 49, 1-47.
>
> [5] Lazaridou, A., Bruni, E., & Baroni, M. (2014, June). Is this a wampimuk? cross-modal mapping between distributional semantics and the visual world. In Proceedings of the 52nd Annual Meeting of the Association for Computational Linguistics (Volume 1: Long Papers) (pp. 1403-1414).
>
> [6] Lazaridou, A., Pham, N. T., & Baroni, M. (2015). Combining language and vision with a multimodal skip-gram model. arXiv preprint arXiv:1501.02598.
>
> [7] Chrupała, G., Kádár, A., & Alishahi, A. (2015). Learning language through pictures. arXiv preprint arXiv:1506.03694.
>
> [8] Kiros, J., Chan, W., & Hinton, G. (2018, July). Illustrative language understanding: Large-scale visual grounding with image search. In Proceedings of the 56th Annual Meeting of the Association for Computational Linguistics (Volume 1: Long Papers) (pp. 922-933).
>
> [9] Zablocki, E., Piwowarski, B., Soulier, L., & Gallinari, P. (2018, April). Learning multi-modal word representation grounded in visual context. In Proceedings of the AAAI Conference on Artificial Intelligence (Vol. 32, No. 1).
>
> [10] Ailem, M., Zhang, B., Bellet, A., Denis, P., & Sha, F. (2018). A probabilistic model for joint learning of word embeddings from texts and images. In Proceedings of the 2018 Conference on Empirical Methods in Natural Language Processing (pp. 1478-1487).
>
> [11] Gupta, T., Schwing, A., & Hoiem, D. (2019). Vico: Word embeddings from visual co-occurrences. In Proceedings of the IEEE/CVF International Conference on Computer Vision (pp. 7425-7434).
>
> [12] Bordes, P., Zablocki, E., Soulier, L., Piwowarski, B., & Gallinari, P. (2020). Incorporating visual semantics into sentence representations within a grounded space. arXiv preprint arXiv:2002.02734.
>
> [13] Vaswani, A., Shazeer, N., Parmar, N., Uszkoreit, J., Jones, L., Gomez, A. N., ... & Polosukhin, I. (2017). Attention is all you need. In Advances in neural information processing systems (pp. 5998-6008).
>
> [14] Devlin, J., Chang, M. W., Lee, K., & Toutanova, K. (2018). Bert: Pre-training of deep bidirectional transformers for language understanding. arXiv preprint arXiv:1810.04805.
>
> [15] Radford, A., Kim, J. W., Hallacy, C., Ramesh, A., Goh, G., Agarwal, S., ... & Sutskever, I. (2021). Learning transferable visual models from natural language supervision. arXiv preprint arXiv:2103.00020.
>
> [16] Jia, C., Yang, Y., Xia, Y., Chen, Y. T., Parekh, Z., Pham, H., ... & Duerig, T. (2021). Scaling up visual and vision-language representation learning with noisy text supervision. arXiv preprint arXiv:2102.05918.

---

### Official Review · Reviewer_Zh5u · 2021-07-17

**Rating:** 7
**Confidence:** 5

**Summary:**

This paper proposes a contrastive learning framework to learn visually grounded, and as a result, semantically interpretable language representations. It proposes two contrastive losses which make use of multiheaded cross-modal attention and bilinear interaction to compute similarity scores between subject and object representations, which can then we used in the contrastive loss.

They perform thorough experiments with the resulting representation to illustrate how well they correlate to human ratings along different language axes such as concrete-abstract and compositionality, as well as qualitative multimodal image search examples.

**Limitations And Societal Impact:**

The authors have not fully addressed limitations of their work. The method, motivation, and qualitative results are important alone, but the goal in having visually grounded and interpretable language representations is to then also use them in multimodal tasks (and replace current representations which are not grounded nor interpretable). The hope is that the representations proposed in this work perform as well as current state of the art or better, but this isn't demonstrated with any downstream experiments. E.g., the multimodal image search section provides a very cool illustration (Figure 7), but then no quantitative experiments on retrieval are provided. Of course the main contribution of this paper concerns interpretability, but connecting the work to real world applications would be an important point to include in the discussion.

As far as societal impact, this paper only provides positive impact, which is great to see!

**Main Review:**

This paper provides a meaningful contribution that naturally follows recent image-text representations learned with a contrastive loss (e.g. CLIP/ALIGN). The writing and structure of the paper is strong and clear, it was easy to read. The authors also clearly motivate the work, as having more semantically meaningful language representations is something very desirable, especially from an ethical and interpretability perspective.

The overall approach and results are well illustrated, I think they could be reproduced fairly easily (although time consuming if associated code is not released). There are a comprehensive and interesting set of experiments on the language representations, which include analyzing representations with PCA and their relationship to SemCat words, use for predicting binary semantic features, how well they cluster according to SemCat categories, how they capture compositional relationships, and how search results change along an image to text feature continuum.

There are some remaining questions I hope can be answered:
1. In Section 4.1, how many principal components are there?
2. L173-175, Bert is discussed as comparison. What about vision-language transformers or other vision-language language models? It seems like the comparison is a bit incomplete/unfair given that there are vl language representations out there which may have better captured the semantic space than Bert (e.g., ViLBERT, LXMERT, VisualBERT, Oscar, MT-GrOVLE etc)
3. The illustration of the significance levels in Figure 4 are pretty hard to follow - why are there overlapping brackets with different *? Shouldn't each box plot be associated with only one of the significance values?
4. Typo at L225, I believe "shape" should be "sharp"

**Time Spent Reviewing:**

2 hours

---

> ### Author Response · Authors · 2021-08-10
> **Response to Reviewer Zh5u**
>
> We are thankful for the reviewer’s summary, encouraging comments, and valid critiques. As the reviewer recognizes, the primary motivation of our work is to evaluate and improve the interpretability of language representations before and after grounding language learning to vision. To us as neuroscientists, this motivation is important from the perspective of cognitive science and machine learning.
>
> We advocate and support reproducible scientific research and commit to sharing our code upon the acceptance of this paper.
>
> As the reviewer points out, our work could benefit from more quantitative reporting and more downstream vision-language tasks. We admit the limitations of our work, in part because our limited computational and data resources precluded us from otherwise leveraging much larger datasets for model training. Unlike the datasets used in related work: CLIP [1] and ALIGN [2], the datasets used in our work are orders of magnitude smaller. For this reason, we focus this paper on characterization of visually grounded language representations, while deferring to our future studies to scale up the model training using the same datasets as for CLIP and ALIGN. Until they are all trained with the same or similarly sized dataset, it is not fair to compare our model with the state-of-the-art models in more downstream applications, e.g., visual question answering, multimodal retrieval. That said, we believe our paper is comprehensive (and partly quantitative) in evaluating the stand-alone language model after visual grounding, as also recognized by the reviewer. Our model design and evaluation yield compelling evidence for interpretable and continuous textual and visual semantics, although some results are illustrative and qualitative and thus limited.
>
> ---
>
> As below we elaborate our detailed replies to other specific comments from the reviewer.
>
> 1. Re the number of principal components, we have applied PCA to the 768-dimensional word embeddings of 9197 common English words in the SemCat dataset. After visual grounding, we focus on analyses and evaluation on the first 6 principal components. The first principal component is highly correlated with the concrete-abstract attribute (rated by humans) and explains 14.1% variance of the grounded word embeddings. The first 167 principal components (that is, 21.7% of the total feature dimension) explain >90% variance. In contrast, the first principal component of the ungrounded word embeddings is not readily interpretable by any human intuition or rating. It requires significantly more components to explain the same amount of variance.
>
> 2. Re the comparison with other vision-language models: thanks for the question. Indeed, it could have been ideal to evaluate our grounded model against other vision-language models. However, most vision-language models do not use a detachable stand-alone language model, because they fuse visual and textual information early in the processing hierarchy. This is unlike our model that the language stream refined by visual grounding can be applied to textual data without accompanying visual input. The prior models most related to our model are CLIP and ALIGN, which both use detachable language and visual streams and contrastive learning similar to our work. Although it has not been tested, either by us or others, we expect that CLIP and ALIGN might also yield grounded semantic representations more interpretable and clusterable than ungrounded counterparts, similar to what we have demonstrated with our model. In our future studies, we will compare our model with CLIP or ALIGN, after our model is also trained with a dataset of a similar size. Currently, such a comparison would not be fair, since our model is trained with orders of magnitude less data (ours: 118K, CLIP: 400M, ALIGN: 1Billion in terms of the number of image-text pairs). Despite the use of much less data, our visually grounded language representations can already align its first PC to human-rated word concreteness with a very high correlation (r=0.87). We will direct our future studies to evaluate and compare our models with VilBert [3], LXMERT [4], and MT-GrOVLE [5], which are further different from our model, because they use different architectures, learning objectives, and are not always separable into stand-alone visual and language streams. Again, we would like to highlight the importance of using the same training dataset to evaluate and compare vision-language models in order to assure fair comparisons.
>
> 3. Re significance levels in Figure 4: Figure 4 is for pairwise comparisons between different language models in terms of the degree of using the resulting language representations to predict human-defined semantic norms. As such, it reflects the comparison of interpretability. More specifically, each box plot shows the predictability (as measured by F1 scores) over a set of semantic norms (e.g., has_wheels) for a specific feature type (e.g., visual perceptual). Each box shows the lower (25%) percentile, the higher (75%) percentile, and the median of F1 scores. Different box color refers to a distinct language model as labeled in the figure legend. We then pairwisely compare the F1-score for each semantic norm across language models with different levels of visual grounding and test the statistical significance with a one-sided Wilcoxon Signed Rank Test, a non-parametric statistical test for matched samples. Thus, each significant value shows whether one language model (with a higher grounding level; on the right side of the bracket) has significantly better prediction performance than the other model (on the left side of the bracket). For example, we found the grounded model is significantly more predictive than Bert on all types of semantic norms except “other perceptual” (the bottom left bracket), while after relational grounding, the model shows better performance than both Bert (the top bracket) and the grounded model (the bottom right bracket). More details about how we conduct this experiment and the related statistical test are described in the supplementary material (Appendix B.2) due to the page limit of the main text. We hope this explanation helps resolve the confusion.
>
> 4. Re the typo, thank you and we will correct it.
>
> ---
> According to the reviewer’s comments, we will further extend the discussion section of this paper to admit and discuss the limitations of our work, tentatively as below.
>
> - Several limitations of our work are noteworthy. The datasets used to train our model are orders of magnitude smaller than those used in recent studies [1,2]. Scaling up the model training with increasingly larger datasets is expected to greatly improve the model’s performance for cross-modal tasks, while generally preserving the interpretability of the grounded language representations as described herein. Some of our experiments and results are preliminary for illustrative purposes and await more comprehensive and quantitative evaluation in future studies, especially with more downstream vision-language tasks. Whereas our evaluation focuses on the language model, grounding language to vision may also have refined the visual stream, awaiting further evaluation against visual tasks, as demonstrated elsewhere [1,2].
>
> ---
>
> References:
>
> [1] Radford, A., Kim, J. W., Hallacy, C., Ramesh, A., Goh, G., Agarwal, S., ... & Sutskever, I. (2021). Learning transferable visual models from natural language supervision. arXiv preprint arXiv:2103.00020.
>
> [2] Jia, C., Yang, Y., Xia, Y., Chen, Y. T., Parekh, Z., Pham, H., ... & Duerig, T. (2021). Scaling up visual and vision-language representation learning with noisy text supervision. arXiv preprint arXiv:2102.05918.
>
> [3] Lu, J., Batra, D., Parikh, D., & Lee, S. (2019). Vilbert: Pretraining task-agnostic visiolinguistic representations for vision-and-language tasks. arXiv preprint arXiv:1908.02265.
>
> [4] Tan, H., & Bansal, M. (2019). Lxmert: Learning cross-modality encoder representations from transformers. arXiv preprint arXiv:1908.07490.
>
> [5] Burns, A., Tan, R., Saenko, K., Sclaroff, S., & Plummer, B. A. (2019). Language features matter: Effective language representations for vision-language tasks. In Proceedings of the IEEE/CVF International Conference on Computer Vision (pp. 7474-7483).

---

> > ### Comment · Reviewer_Zh5u · 2021-08-25
> > **Response**
> >
> > Thanks for your response. After reading other reviewers critiques, I think it is important to revise the writing to be more clear about the contributions of the paper (i.e. incorporating some of the clarifications made in the rebuttal).
> >
> > I will be updating my score to a 7.

---

> > > ### Author Response · Authors · 2021-08-25
> > > **Thank you for your reply**
> > >
> > > Thank you for your reply. We respect the reviewer’s updated rating. We will revise the paper and clearly state our contributions based on the suggestions from the reviewers.

---

### Official Review · Reviewer_1xyN · 2021-07-18

**Rating:** 7
**Confidence:** 4

**Summary:**

This paper analyzes the linguistic embedding in the joint image-language space learned by cross-modal contrastive learning. Comparing with the standard word embedding, the grounded language embedding is human explainable. Specifically, the PCA of the grounded langauge embedding is aligned with human rating. When clustering the language embedding, the cluster of grounded language embedding is more semantically coherent than the un-grounded language embedding. Furthermore, the paper showed that the grounded langauge embedding could benefit visual-linguistic downstream tasks.

**Limitations And Societal Impact:**

Yes

**Main Review:**

Novelty:

1. Method: The main focus of this paper is to show that the grounded linguistic embedding is more aligned with human perception. Therefore, in terms of the method, the paper uses a pretty standard two stream vision and language model to learn the embedding. The description of the approach is clean and easy to follow.

2. Experiment:the analysis conducted in this paper is pretty interesting. However, this paper is not the first one to conduct similar experiments.

  2.a In terms of 'Clustering of word representations', ViCo (Gupta et, al., 2019) conducted similar experiments (Sect. 4.1) and showed the grounded embedding is better than the un-grounded.

  2.b In terms of the Multimodal image search, Ailem et. al, 2018 conducted similar experiments. They showed that the grounded embedding can help the image retrieval model to achieve better performance than the un-grounded.

Therefore, the experiments and analysis are very interesting. However, they are not surprising given the previous research.

---
Experiment:

1. Word coverage rate: The experiments in Sect. 4.1 and 4.4 are very interesting. However, I wonder whether all the tokens in those two datasets are covered by MS-COCO? Specifically, MS-COCO is annotated with image and text. The text is descriptive sentence which describe the object/event in the image. In Sect. 4.1 the dataset contains word like 'emotions' and 'mythical_beasts' (Fig. 3). I wonder whether those words are covered by the MS-COCO or Visual Genome?

2. In Sect. 4.1, does the ungrounded word embedding show similar behavior? Specifically, could the PCA on the ungrounded word embedding provide any useful interpretation?

3. Sect. 4.5. I am not quite sure how did the image-text retrieval conduct? I could understand everything till L236. But I don't know hwy a query image and a query text need to be combined. How to use the combined representation to conduct image retrieval.

4. Sect. 4.5. As the MS-COCO is a dataset defined for image retrieval, I wonder why the standard image retrieval task and metrics (Recall) is not used/reported?

---
The rating is based on the novelty and the questions in experiment. I would happy to raise the score if the concerns are resolved.

Gupta T. et. al., ViCo: Word Embeddings from Visual Co-occurrences, ICCV 2019.
Ailem, M. et. al., A Probabilistic Model for Joint Learning of Word Embeddings from Texts and Images. EMNLP 2018.

**Time Spent Reviewing:**

6

---

> ### Author Response · Authors · 2021-08-10
> **Response to Reviewer 1xyN**
>
> Thanks for the concise summary. It is encouraging that the reviewer has found and commented that our experiments, analyses, and results are very interesting. In this work, our intuition and working hypothesis is that visual grounding enriches and improves the interpretability of language representations compared to ungrounded counterparts learned from texts alone. We do not claim ourselves to be the first who have ever raised and tested this hypothesis. To the contrary, we admit, and actually emphasize, that language grounding to perception and action has been a long-standing topic in computational linguistics, cognitive science, and more recently machine learning. However, we believe that grounded language models are not yet the common place or default choices in natural language processing. Language models are still mostly learned from texts alone. Vision-language models are not always separable into stand-alone language and vision models that can be detached and applied to language and visual input respectively. The characterization of grounded semantics in relation to human ratings has been incomplete. In the context of literature, we would like to highlight the novelty of our work as below.
>
> - Our model applies visual grounding to not only words but also relations between words based on contrastive learning. To the best of our knowledge, relational grounding has not been explored in prior work and is thus a unique contribution from our work. Contrastive learning has been used in two related papers [1,2] but is still a relatively under-explored strategy for vision-language learning.
> - Our paper reports a novel finding that visually grounded language representations exhibit several principal dimensions that are intuitively interpretable. For example, the first principal component is highly correlated (0.87) with human-related concreteness.
> - Our paper also shows compelling evidence that both text and image-informed semantics are represented in a common, continuous, and grounded semantic space. This is a notion that has been advocated and hypothesized in neuroscience and linguistics; however, it has not been rarely demonstrated with computational models. In this paper, we demonstrate this in a unique yet illustrative experiment that a continuously varying combination of a text and an image into a multimodal query can be used to search images, showing results that make intuitive sense.
>
> However, our intention is not to overly claim novelty. Instead, we are happy to acknowledge that our work is in line with related prior work in the field of computational linguistics, cognitive science, and machine learning. In a future revision of this paper, we will extend our discussion about the related work, tentatively as below.
>
> Grounding language in vision has been of increasing interest in computational linguistics and machine learning. A common strategy is to fuse words with related visual information in terms of perceptual norms [3], bag-of-visual-word [4,5], or learnable features [6,7,8,10]. The models used for vision-language fusion evolve alongside those for natural language processing, such as Latent Dirichlet Allocation (LDA) [4], log-bilinear model (like GloVe) [11], Skip-gram model (like word2vec) [6,9], recurrent neural network [7]. Findings from prior work suggest that visual grounding may refine the distribution and interpretability of language representations [4,5,7,8,9,10,11,12] and facilitate cross-modal tasks [7,8,10,11,12].
>
> ---
>
> Answers to detailed comments:
>
> 1. Re the word coverage, we have investigated the word coverage rate in MS COCO dataset. The experiment in Sect. 4.1 is based on the SemCat dataset. 136 words in the ‘emotions’ category appear in the training dataset for 67.22 ± 47.65 times. 50 words in the ‘mythical_beasts’ category appear in the training dataset for 22.12 ± 15.98 times (mean ± standard error). The most frequently sampled word categories are 'people' (1865.72 ± 742.23), 'colors' (1612.16 ± 534.19), 'postal', 'kitchen', 'furniture', 'restaurant', 'many', 'baseball', 'circus', 'driving'. The least frequently sampled word categories are 'sciences' (0.30 ± 0.13, for words like biology, kinetics, physics etc.), 'country' (4.13 ± 2.00), 'reptiles', 'languages', 'virtues', 'mythical_beasts', 'grammar', 'election', 'doctor', 'money'. This follows our expectation since the text inputs in the training data are all descriptions of natural images. However, we have also found the word occurrence rate in the training dataset is not correlated with the clustering performance shown in Sect. 4.3 (see Fig. 13 in the supplementary materials). For the experiment in Sect. 4.4, most of the words contain rich visual features and have been sampled frequently in the training dataset (word:occurrence): e.g., zebra:4890, panda:163, plane:4431, bowl:4634, strawberry:97, puppy:281, glacier:4, sunny:1340, summer:65). We will add the information regarding word coverage rate to the Appendix in our revised version.
>
> 2. Re the PCA of ungrounded word embeddings. No, the ungrounded word embedding space from pre-trained Bert does not show similar behaviors as the grounded one from our model. The first PC from ungrounded semantic space only reached 0.35 correlation (see line 174 in the main text and Fig. 7 in the supplementary material) with human-rated word concreteness. We have further looked into the first 10 principal components, and none of them showed a strong correlation or appeared to be readily interpretable. In contrast, after grounding, the first principal component from the grounded word embedding space is correlated with human rated concreteness with r=0.87. The second and third principal components also provide intuitive and useful interpretation.
>
> 3. Re image-text retrieval or multimodal image search, we will extend our description to enhance clarity. Briefly, we first average-pool the output from the visual stream (thus the image feature shrinks from a 3D tensor with size H x W x N_emb to a 1D vector $Q_I \in \mathbb{R}^{N_{emb}}$) and the output from the language stream (thus the word or phrase input is also represented by a 1D vector $Q_W \in \mathbb{R}^{N_{emb}}$ with the same dimension). These two representations are L2 normalized and aligned into a shared space through contrastive learning. Then to construct a search query ($Q_{search}$), we use a linear combination of a pair of image query ($Q_I$) and text query ($Q_W$) (since they are now in the same representational space) weighted by a scalar $\alpha \in [0,1]$. When $\alpha=0$, only the image is used as a query for search and retrieval. When $\alpha=1$, only the word is used as the search query. And when $\alpha=0.5$, the image and text inputs contribute equally to the search query. The weight $\alpha$ can vary continuously in the range from 0 to 1. If the visual and text-informed semantics share a common semantic space, then varying alpha is expected to result in the retrieved images to vary their contents according to progressive and intuitive transition from the image to the text.  This hypothesis is supported by the fact that we are able to see a “continuous” transition in the image search result shown in Figure 7, from zebra skin pattern (image only), to a real zebra (image+text), to a horse (text only). We hope our explanation helps resolve the confusion.
>
> 4. Re the standard image retrieval tasks, we agree that it seems natural to conduct a standard image-text retrieval task after training our model on MS COCO. We have reported the results in the supplementary material (see Appendix A.2 and Fig. 3). However, we didn’t include this in the main text due to the page limit. The retrieval performance we achieved, e.g. Grounded-8 has 25.3% (I2T) and 23.9% (T2I) accuracy, was lower than the state-of-the-art mainly because we use a relatively small training dataset (118K image-caption pairs), compared to VilBERT [13] (Conceptual Captions dataset contains 3.3M image-caption pairs), Oscar [14] (6.5M image-text pairs), CLIP [1] (400M image-caption pairs), ALIGN [2] (1.8B image-text pairs).

---

> > ### Author Response · Authors · 2021-08-10
> > **Response to Reviewer 1xyN (references)**
> >
> > References:
> >
> > [1] Radford, A., Kim, J. W., Hallacy, C., Ramesh, A., Goh, G., Agarwal, S., ... & Sutskever, I. (2021). Learning transferable visual models from natural language supervision. arXiv preprint arXiv:2103.00020.
> >
> > [2] Jia, C., Yang, Y., Xia, Y., Chen, Y. T., Parekh, Z., Pham, H., ... & Duerig, T. (2021). Scaling up visual and vision-language representation learning with noisy text supervision. arXiv preprint arXiv:2102.05918.
> >
> > [3] Silberer, C., & Lapata, M. (2012, July). Grounded models of semantic representation. In Proceedings of the 2012 Joint Conference on Empirical Methods in Natural Language Processing and Computational Natural Language Learning (pp. 1423-1433).
> >
> > [4] Bruni, E., Tran, N. K., & Baroni, M. (2014). Multimodal distributional semantics. Journal of artificial intelligence research, 49, 1-47.
> >
> > [5] Lazaridou, A., Bruni, E., & Baroni, M. (2014, June). Is this a wampimuk? cross-modal mapping between distributional semantics and the visual world. In Proceedings of the 52nd Annual Meeting of the Association for Computational Linguistics (Volume 1: Long Papers) (pp. 1403-1414).
> >
> > [6] Lazaridou, A., Pham, N. T., & Baroni, M. (2015). Combining language and vision with a multimodal skip-gram model. arXiv preprint arXiv:1501.02598.
> >
> > [7] Chrupała, G., Kádár, A., & Alishahi, A. (2015). Learning language through pictures. arXiv preprint arXiv:1506.03694.
> >
> > [8] Kiros, J., Chan, W., & Hinton, G. (2018, July). Illustrative language understanding: Large-scale visual grounding with image search. In Proceedings of the 56th Annual Meeting of the Association for Computational Linguistics (Volume 1: Long Papers) (pp. 922-933).
> >
> > [9] Zablocki, E., Piwowarski, B., Soulier, L., & Gallinari, P. (2018, April). Learning multi-modal word representation grounded in visual context. In Proceedings of the AAAI Conference on Artificial Intelligence (Vol. 32, No. 1).
> >
> > [10] Ailem, M., Zhang, B., Bellet, A., Denis, P., & Sha, F. (2018). A probabilistic model for joint learning of word embeddings from texts and images. In Proceedings of the 2018 Conference on Empirical Methods in Natural Language Processing (pp. 1478-1487).
> >
> > [11] Gupta, T., Schwing, A., & Hoiem, D. (2019). Vico: Word embeddings from visual co-occurrences. In Proceedings of the IEEE/CVF International Conference on Computer Vision (pp. 7425-7434).
> >
> > [12] Bordes, P., Zablocki, E., Soulier, L., Piwowarski, B., & Gallinari, P. (2020). Incorporating visual semantics into sentence representations within a grounded space. arXiv preprint arXiv:2002.02734.
> >
> > [13] Lu, J., Batra, D., Parikh, D., & Lee, S. (2019). Vilbert: Pretraining task-agnostic visiolinguistic representations for vision-and-language tasks. arXiv preprint arXiv:1908.02265.
> >
> > [14] Li, X., Yin, X., Li, C., Zhang, P., Hu, X., Zhang, L., ... & Gao, J. (2020, August). Oscar: Object-semantics aligned pre-training for vision-language tasks. In European Conference on Computer Vision (pp. 121-137). Springer, Cham.

---

> ### Comment · Reviewer_1xyN · 2021-09-02
> **Updates after the rebuttal**
>
> Thanks for the detailed rebuttal. My major conerns in the first round is the novelty and the unclear part of the experiments.
>
> After the rebuttal I think this paper is a pretty good analysis paper for visual grounded word representation. As this paper provides experiments/evidences to support 1. visual grounded word representation is more aligned with human perception, and 2. image and text lives in the same joint space. Actually the multimodal image search is pretty interesting. I would raise my score to 7, if the author provides more implementation details to the Sect. 4.5.

---

> > ### Author Response · Authors · 2021-09-02
> > **Thank you for your feedback.**
> >
> > Thank you for your constructive and encouraging feedback. We will revise the paper to further clarify the detailed method used for multimodal image search in Section 4.5. In addition, we will also describe more implementation details and include a script for running multimodal image search examples in the supplementary material.

---

### Official Review · Reviewer_BB9g · 2021-07-21

**Rating:** 5
**Confidence:** 4

**Summary:**

The paper aims to ground language to vision via cross-modal contrastive learning to build an explainable shared semantic space. This is achieved in two parts (i) align visual and language representations with MS COCO dataset (ii) Retrieve visual objects with language queries through a cross-modal attention module and infer the visual relations between the retrieved objects through a bilinear operator on Visual Genome dataset. This allows model's language encoder to become a stand-alone language model that can encoder embedding concepts in a visually grounded semantic space. The model enables compositional language understanding based on visual knowledge and multimodal image search with queries. The authors show experiments on clustering, compositional reasoning, relationship with human-defined norms, and image search to showcase the usefulness of the model.

**Limitations And Societal Impact:**

The paper doesn't have a broader impact statement included.

**Main Review:**

The paper targets a less-studied problem of grounding language in visual semantic space and uses a two-step process to achieve that. Specifically, for the first step the paper uses a CLIP-style [1] contrastive loss to align visual and language representations using MS COCO dataset. For the second step, which seems useful in helping ground the object relations to generate visually ground object representation for the text, the model uses a cross-attention module similar to VILBERT [2] model, by passing language stream as a query and visual output as a key and value to generate a language conditioned visual representation which is basically visually grounded object representation for the object present in the text. A bilinear relation module is applied to the ground object representation for predicting the relationship between the two objects which is trained using a contrastive learning approach. The idea of aligning using MS COCO dataset uses CLIP-style contrastive loss and is not new while the idea of improving the object semantic grounding using Visual Genome dataset and contrastive loss over relations is new and insightful. Comments on the paper are as follows:

1) The paper doesn't provide a concrete way of evaluating the usefulness of these embeddings on the downstream vision-and-language tasks in a quantitative way. I was expecting to see some experiments similar to CLIP paper on how exactly the better grounding can help achieve a better general model for downstream tasks. The paper does cover a qualitative study on multimodal image search but didn't do extensive experiments on image or text retrieval which this model would have empowered.The supplementary material provides some results on retrieval on MS-COCO dataset but the number seem to very low compared to state-of-the-art model which seems surprising (the appendix only have qualitative figure for comparison and no table). This leaves the reader with incomplete knowledge on the actual practical usefulness of the model.
2) The section 4.3 is complicated and complex to understand in the way it is written. I would suggest that authors to either improving the writing or remove that section for the sake of the clarity of the paper. It took me some time to understand how figure 6 was working and what it was supposed to show after reading section 4.3.
3) The paper does evaluate self-attention on top of VGG16 feature extractor they use for imagenet classification accuracy (in appendix) but never evaluate their final visual stream on imagenet to check how grounding helps.
4) The paper also evaluates on GLUE tasks in appendix but it looks like the performance is decreasing on most tasks which is counter-intuitive as learning about world through images should maintain or improve original BERT model. I would suggest that authors take a look at [3] which aims to learn multimodal and unimodal tasks using a single transformer architecture.
5) The paper doesn't provide any ablations on not using "visual grounding of object relations" stage or only using that stage. These ablations are necessary and insightful to get a sense to how much visual grounding of object relations is helping the model in comparison to just using CLIP-style loss. Ideally, if it helps, it could also help improve the already strong CLIP model as well. Furthermore, the ablation study on only using "visual grounding of object relations" could help understand how much CLIP-style loss helps in overall grounding of the language to vision and vice versa.
6) Similarly, the ablations of Loss_l and Loss_v alone are missing to understand impact of two modalities on contrastive loss. The paper also doesn't provide ablations on how much data helps in how much performance. Ideally, a study could be conducted on using percentage of coco dataset to showcase if more data can help in the grounding process. The paper also doesn't mention any hyperparameter tuning or otherwise.
7) There is a lot going on in the appendix and some of the things there can make it into the paper (such as ablations on visual grounding loss's components relations and object prediction loss) while some the things like I mentioned can be moved to supplementary.

The paper is a good attempt in a less-explored direction but fails to provide reader with useful insights on actual practical help of doing the visual grounding as GLUE results drop, multimodal retrieval results are not impressive among others. Though qualitative results make sense around clustering and probing, the model at large doesn't look useful from the provided information in the manuscript for the downstream tasks.

[1] Radford, Alec, et al. "Learning transferable visual models from natural language supervision." arXiv preprint arXiv:2103.00020 (2021).

[2] Lu, Jiasen, et al. "Vilbert: Pretraining task-agnostic visiolinguistic representations for vision-and-language tasks." arXiv preprint arXiv:1908.02265 (2019).

[3] Hu, Ronghang, and Amanpreet Singh. "UniT: Multimodal Multitask Learning with a Unified Transformer." arXiv preprint arXiv:2102.10772 (2021).

**Time Spent Reviewing:**

7

---

> ### Author Response · Authors · 2021-08-10
> **Response to Reviewer BB9g (part 1)**
>
> Thanks for the summary. We respect the reviewer’s comments and opinions. We do not intend to push the model’s performance in the context of cross-modal tasks; instead, the primary purpose of our work is to explore ways to ground language to vision and evaluate the interpretability of the grounded language representations.
>
> There are multiple novel aspects about our work: 1) visual grounding of object relations as well as words/sentences through contrastive learning, 2) the principal axis of the grounded semantic space reflecting the human-rated concreteness, 3) text and image informed semantics represented in a common, continuous, and grounded semantic space. We believe that these aspects, collectively, make solid contributions to language grounding to vision, which is of common interest to computational linguistics, cognitive science, and machine learning.
>
> That said, the reviewer’s critiques are valid. We agree that lacking systematic evaluations on downstream tasks is a limitation of the current study. There are two major reasons why we chose to focus on the experiments and results reported in the current paper. First, the research question we want to pursue is to understand how visual grounding affects the language processing stream and the resulting characteristics of the semantic space, which remains a relatively less-studied question in the field. Second, in this preliminary work, we used a relatively small set of data for training (COCO dataset contains 118K image-caption pairs, Visual Genome dataset after cleaning contains 94K images), compared to VilBERT [1] (Conceptual Captions dataset contains 3.3M image-caption pairs), Oscar [2] (6.5M image-text pairs), CLIP [3] (400M image-caption pairs), ALIGN [4] (1.8B image-text pairs), etc. Thus, we did not aim to or expect to achieve comparable performance as these prior models trained with datasets that are orders of magnitude larger. However, results from our experiments have demonstrated that visual grounding of transformer-based language models enables interpretable language representations than its ungrounded counterpart. In future work, we would like to investigate and evaluate the performance on multiple downstream tasks after scaling up the current model by training it with a larger dataset. Currently we are short of computational resources to do that.
>
> We would like to elaborate our rationale in the following replies to specific comments from the reviewer. Hopefully, the reviewer would be amenable to our motivation and interest.
>
> ---
>
> Answers to detailed comments:
>
> 1. We agree that it would be ideal if the model is evaluated for many downstream tasks. However, the model’s performance on downstream vision-language tasks is really not the focus of this paper. The research question that we set out to pursue is to understand how visual grounding affects language representations with a special emphasis on their interpretability. With respect to this question of interest, our manuscript has reported a large set of experiments and evaluations. To limit the scope of this paper to our primary focus, we would like to defer further model evaluation with downstream cross-modal tasks, after we scale up the model training with larger datasets. Given the constraints of our computational and data resources, our model is trained only with a relatively small dataset (118K image-caption pairs). The amount of training data used in our work is orders of magnitude less than the data used in related recent work. For example, ALIGN [4] is trained with 1.8B image-text pairs and then is able to achieve the state-of-the-art image-text retrieval performance on the MS COCO 5K testing dataset. In their ablation studies, the authors of that paper have shown that the retrieval performance would drop, significantly, to 18.9% (I2T) and 15.5% (T2I) if their model is trained on a smaller dataset: Conceptual Captions (3M), which is still >15 times larger than the size of our training dataset. In contrast, our model is able to achieve a reasonably good retrieval performance, e.g., 25.3% (I2T) and 23.9% (T2I) accuracy, even though we only use much less training data (118K). Besides, evidence from prior studies [5,6] also suggests that although contrastive learning objectives can learn better representations than predictive/classification objectives, the model trained with contrastive learning may perform worse on image-text retrieval than those trained explicitly for cross-modal retrieval (like binary classification loss). Thus, it is reasonable that our model shows lower performance than the state-of-the-art on this task. Figure 3 in the supplementary material shows the effect of learnable layers and/or attention parameters on the performance of cross-modal retrieval. According to the reviewer’s comment (“no table”), we will add a table to show the exact numbers of the retrieval accuracy in our revised version.
>
> 2. We are regretful that Figure 6 does not appear to be straightforward to understand. The figure demonstrates a very nice clustering behavior and arguably a compelling finding. However, we are restricted by the page limit and a large set of experiments and results to report and we do not have enough space to further extend the related description. To reconcile with the reviewer’s comment and suggestion, we will move Figure 6 to the supplementary material.
>
> 3. We agree that it would be better to evaluate both the language model and the vision model after grounding. However, we choose to focus this paper on the language model, because 1) that is our primary research question of interest, and 2) prior work, CLIP [3] and ALIGN [4] published earlier this year, focuses their attention on the vision model after pretraining with “natural language supervision”. Although our work shares some similar methodological choices, our perspective is complementary to the prior work, focusing on how “visual supervision” with contrastive learning changes the language stream. Given our focus on the language representation, our results suggest that visual grounding enables a more interpretable semantic space that aligns with human intuition and neurobiological knowledge. Hopefully, the reviewer is willing to be amenable to our intention to stay on track along our research interest, complementary to literature. Our work, as well as the prior work [3, 4], altogether present coherent evidence to support the use of contrastive learning for vision-language representation learning.
>
> 4. Unlike the reviewer’s expectation, we do not expect visual grounding to improve GLUE tasks, given the model as implemented and trained with the COCO and Visual Genome datasets. Instead, we would like to clarify that our motivation for evaluating the GLUE benchmark is to test whether fine-tuning BERT on COCO and Visual Genome dataset may compromise the performance of the language stream on textual processing. This is a fair concern since the text inputs from these two datasets are either short sentences or phrases, which are very different from the BERT pretraining that used long textual sequences. Thus, although the output language representations have shown promising interpretable properties, they might lose some capability for natural language processing over a much broader textual context. Besides, for this specific evaluation, we have fixed the language encoder after model training and only re-trained an add-on task-specific linear classifier, which can be viewed as a simple transfer learning evaluation of the learned language representation. Our motivation and approach are very different from the UniT [7] paper, where the whole model (including visual encoder, text encoder, and a joint decoder) is trained on multiple unimodal and multimodal tasks, including four of the GLUE tasks. Although being different from the weak-supervised pretraining setting in our work, UniT [7] indeed demonstrates an intriguing idea to train a unified transformer with multiple cross-modal tasks for learning multimodal representation. We are glad that the reviewer refers us to this prior work.
>
> 5. We choose not to use relational grounding to train the model, while skipping the step of image-text alignment. This is because of the limited dataset usable for grounding subject-object relations. Scaling up the relational grounding with a much larger dataset is expected to be more effective for refining the grounded language model and reshaping the semantic space. Given the restricted training data, relational grounding offers relatively minor refinement compared to its preceding stage. This is likely because the stage of “visual grounding of natural language” has already aligned the visual and language representations quite well, while the following stage of “visual grounding of object relations” provides more details on object-level information to refine that space.

---

> > ### Author Response · Authors · 2021-08-10
> > **Response to Reviewer BB9g (part 2)**
> >
> > 6. This is an interesting question. Yes, it is worth checking how loss_l and loss_v individually contribute to the contrastive learning process during the visual grounding of natural language. Due to the limited time for the first round of review responses, we have not finished all the ablation experiments. We will add additional results in a revised version. Besides, we agree that it is worth using partial training data to investigate the impact of training size, to validate and compare with the findings in prior works like [4]. We have tuned a subset of the hyperparameters (e.g. learning rate, dropout rate, the temperature parameters in the contrastive loss functions, the learnable layers in the Bert base model, etc.), some related results were reported in the appendix (Fig. 3 and Fig. 12). However, we have not done a thorough search in the hyperparameter space to find the optimal training setting. In addition, we have also found that the findings about the semantic space evaluations reported in Section 4 are largely consistent for different hyperparameters we have tested so far as long as the loss converges.
> >
> > 7. Thank you for this comment. We agree with you that many results that we put in the supplementary materials are as important as the ones in the main text, but we have to balance the content due to the page limit. To narrow down the scope and keep the focus of the current work, we prefer to have the results that are most relevant to demonstrating the explainability of the semantic space after visual grounding in the main text. That said, if the paper is accepted, we will try to adjust the final version according to the reviewer’s suggestions.
> >
> > ---
> >
> > References:
> >
> > [1] Lu, J., Batra, D., Parikh, D., & Lee, S. (2019). Vilbert: Pretraining task-agnostic visiolinguistic representations for vision-and-language tasks. arXiv preprint arXiv:1908.02265.
> >
> > [2] Li, X., Yin, X., Li, C., Zhang, P., Hu, X., Zhang, L., ... & Gao, J. (2020, August). Oscar: Object-semantics aligned pre-training for vision-language tasks. In European Conference on Computer Vision (pp. 121-137). Springer, Cham.
> >
> > [3] Radford, A., Kim, J. W., Hallacy, C., Ramesh, A., Goh, G., Agarwal, S., ... & Sutskever, I. (2021). Learning transferable visual models from natural language supervision. arXiv preprint arXiv:2103.00020.
> >
> > [4] Jia, C., Yang, Y., Xia, Y., Chen, Y. T., Parekh, Z., Pham, H., ... & Duerig, T. (2021). Scaling up visual and vision-language representation learning with noisy text supervision. arXiv preprint arXiv:2102.05918.
> >
> > [5] Tian, Y., Krishnan, D., & Isola, P. (2020). Contrastive multiview coding. In Computer Vision–ECCV 2020: 16th European Conference, Glasgow, UK, August 23–28, 2020, Proceedings, Part XI 16 (pp. 776-794). Springer International Publishing.
> >
> > [6] Qi, D., Su, L., Song, J., Cui, E., Bharti, T., & Sacheti, A. (2020). Imagebert: Cross-modal pre-training with large-scale weak-supervised image-text data. arXiv preprint arXiv:2001.07966.
> >
> > [7] Hu, R., & Singh, A. (2021). UniT: Multimodal Multitask Learning with a Unified Transformer. arXiv preprint arXiv:2102.10772.

---

> > > ### Comment · Reviewer_BB9g · 2021-08-23
> > > **Thanks for the response**
> > >
> > > Thanks for providing a detailed response to my queries. Unfortunately, most of my concerns (downstream tasks, ablations, etc.) still stand though I understand the viewpoint authors are coming from.
> > >
> > > 1. I understand that authors want to scale up and then check the downstream performance but in the current version, the paper isn't showing anything extra compared to what CLIP and ALIGN already show in some sense. I would expected that VG based pretraining would helped in better grounding and an easy way to show that would have been to do proper ablations on downstream tasks to compare these with and without VG pretraining but those are also still missing. I understand the lack of compute but running the ablations and comparisons in a smart way would be possible to show gains which can be extrapolated to CLIP-scale data if available later. I hope authors understand why evaluations are necessary given that CLIP and ALIGN already exist.
> > >
> > > 2. Thanks. I had hoped authors could had given a simpler explanation here.
> > >
> > > 3. Again as 1. I believe proper juxtaposition with the most related work CLIP and ALIGN is necessary and one way to do that is to compare the effectiveness of the visual encoder after pretraining. As I suggested in 1, ablation study can again help with that.
> > >
> > > 4. I understand authors' response but I believe we need to agree to disagree here. I think here also an ablation can be showed by starting from BERT pretrained encoder to see if semantic understanding can help the embeddings further.
> > >
> > > 5. I believe this ablation is important for the holistic understanding of the approach and benefits over CLIP/ALIGN and my original point still stands after reading author response.
> > >
> > > 6. It would have been great if authors had actually included results here. So, this concern also stands as of now.
> > >
> > > Given that the paper doesn't really provide any extra insight on top of CLIP/ALIGN and fails to support the novel contributions with proper ablations and downstream tasks, I would like to keep my initial rating as it is. I once again would like to thank authors for involving in this discussion seriously and writing this paper.

---

> > > > ### Author Response · Authors · 2021-08-25
> > > > **Thank you for your comments**
> > > >
> > > > We would like to thank the reviewer for his/her time reviewing the paper and offering critiques. Despite the limited time for us to address the critiques and run new experiments, we believe our replies are respectful and responsive, clarifying the purpose of the paper, justifying our choice of methods, and highlighting our contributions in the context of a growing body of literature on vision-language learning. However, our attempt to have a respectful, professional, and scientific exchange does not end up with mutual understanding or constructive feedback. If our replies were unsatisfactory or irresponsive to the reviewer’s questions, we hope the reviewer could comment on specifics in our paper or reply to help us understand the reviewer’s critiques. Otherwise, the reviewer’s position to insistently dismiss our intended research questions, experimental results, and scientific justification is regretful and discouraging a fair dialog or an informed peer review.
> > > >
> > > > We respect the reviewer’s right to keep his/her rating of our paper. Nevertheless, we still elaborate our replies given our best understanding of the reviewer’s comments (admittedly we cannot understand some comments from the reviewer because of grammatical errors, incomplete phrasing, or lack of specifics). Hopefully, our replies help clarify misunderstandings.
> > > >
> > > > 1. It is critical for us to state again that the main goal of this paper is to test whether the visual grounding of natural language would give rise to a language model that could yield more interpretable language representations. The experiments and results described in the paper are intended to serve this purpose from different perspectives. Our goal is related to but different from those of the prior work, namely CLIP and ALIGN, both of which focus on pretraining a visual model by vision-language learning. In short, our work focuses on the language model, whereas CLIP and ALIGN focus more on the visual model and visual tasks. We hope the reviewer would be amenable to our intention in writing a paper of our own interest (as the authors). In addition, because we do not have access to the equivalent data or computing power as in CLIP or ALIGN, it is not fair to compare our model with CLIP or ALIGN only with respect to performance with downstream tasks, which also deviate from our primary focus. We respectfully disagree with the reviewer’s assessment: “the paper isn't showing anything extra compared to what CLIP and ALIGN already show in some sense”. Neither CLIP nor ALIGN has addressed the interpretability of visually grounded semantic representations. Our paper addresses this to a considerable depth. Our method also applies visual grounding to object relations. Isn’t that something “extra”? We hope that the reviewer is willing to reconcile his/her comments with the comments from other peer reviewers who have recognized our intellectual contributions. Lastly, we have also explored ablation or variation to the model design (including new experiments per the reviewer’s request). Because our focus is on the interpretability of semantic representations, part of the results less related to this focus are included in supplementary materials. Our evaluation focused on the visually grounded language stream of our model is comprehensive, as shown in both the main text and supplementary material of this paper.
> > > > 2. In Fig.6, we illustrate how words within a category (e.g., vehicle) are represented in the embedding space. In this “vehicle” example, we first take “boat”, “car”, and “airplane” as three “query words”. We define 15 words of interest that belong to each of these three subcategories, i.e. *water/land/air transportation* (totally 45 words, see Table 7 in the supplementary material for a list of words). For example, *yacht* and *canoe* are “boat”-like vehicles, *suv* and *taxi* are “car”-like vehicles, *jet* and *flight* are “airplane”-like vehicles. Then, for each word of interest, we calculate its cosine similarity with the three “query words”, which results in a three-dimensional “representational similarity” vector. These 45 vectors are visualized in three 2D planes (as shown in Fig.6), each spanned by a pair of query words. After color-coding words by their subcategories, we demonstrate that the representations of these word subcategories are more separable and better clustered after visual grounding, suggesting that visual information (e.g. has wheels or has wings) helps to construct better word embeddings in the grounded language model.
> > > > 3. CLIP and ALIGN focus on the visual encoder. Our work focuses on the language encoder and representation, which is a major distinction from CLIP and ALIGN. We hope the reviewer is open and willing to respect our intended goal and interest in this paper. The reviewer and the authors clearly have different perspectives and interests. We hope that the reviewer will allow our scientific interest to be expressed in our own paper. We respect disagreement, especially if the disagreement exists after the difference is scientifically justified. Taking “one’s way or the highway” is counterproductive in peer review.
> > > > 4. We are unsure how to understand the comments: “agree to disagree”, “if semantic understanding can help the embedding further”. The reviewer feels free to disagree; but hopefully the reviewer would be willing to elaborate on the difference. The language models with and without visual grounding have been compared in section 4.1 to section 4.4. Again, our paper is about the effect of visual grounding on language learning and representation, not about the effect of “semantic understanding” on visual learning and representation.
> > > > 5. According to the reviewer’s previous comment, we have also completed a new analysis (as committed in our previous reply) on the ablations of loss$_l$ and loss$_v$ (as defined in equation 2). We have trained the model described in section 3.1 with three different objective functions: a) loss$_l$+loss$_v$ (original); b) loss$_l$ only; c) loss$_v$ only. Our hypothesis is that by combining loss$_l$ and loss$_v$, the model will reach the best performance for cross-modal image-text retrieval. Our results support this hypothesis.
> > > >     > Here is the summarization of the cross-modal retrieval performance (top-1 accuracy) on COCO 5k test set:
> > > >     >   - a) 23.9% (text-to-image), 25.3% (image-to-text)
> > > >     >   - b) 17.4% (text-to-image), 25.5% (image-to-text)
> > > >     >   - c) 25.5% (text-to-image), 1.4% (image-to-text)
> > > >
> > > > In brief, if we only use loss$_v$ (which contrasts between positive and negative image keys) for training, the image-to-text retrieval significantly drops from 25.3% to 1.4%, although the text-to-image retrieval performance increases by 1.6%. On the other hand, if we only use loss$_l$ (which contrasts between positive and negative language keys), the text-to-image retrieval drops by 6.5%, while the image-to-text retrieval just increases by 0.2%. The overall cross-modal retrieval achieves the best performance when both losses are used for model training.
> > > >
> > > > ---
> > > >
> > > > Again, our work is related to but different from CLIP and ALIGN. It sheds new light on the fact that visual grounding with cross-modal contrastive learning leads to a grounded language model and helps learn more interpretable and better clustered language representations. We believe, as recognized by other reviewers, that our findings about interpretable principal dimensions of semantic space after visual grounding are particularly novel and compelling, and arguably insightful from the perspectives of cognitive science, computational linguistics, and NLP.
> > > >
> > > > For a more balanced review process, we will appreciate the oversight from the area chairs and the engagement of other reviewers. Thank you for your time and effort.

---

### Decision · Program_Chairs · 2021-09-27

**Decision:**

Accept (Poster)

**Comment:**

This paper investigates cross-modal contrastive learning for semantic representations and finds that grounded language embeddings are more semantically coherent than un-grounded ones. The reviewers generally like this direction and the paper is well-written. The authors should really heed the feedback from the reviewers, however, and in particular: the authors should make *very* clear that this is not a methods paper, but an analysis paper. Occasionally the paper reads too much like it is claiming a new method, which is problematic because a) the comparison to other models is lacking; and b) the paper is not adequately positioned in the existing literature. There are many papers that have explored visually-grounded semantic embeddings in the past, several of them already quite old, and this work should be positioned accordingly.